# Isolated copper single sites for high-performance electroreduction of carbon monoxide to multicarbon products

Haihong Bao[1,11], Yuan Qiu[1,11], Xianyun Peng[1,11], Jia-ao Wang [2,11], Yuying Mi[1], Shunzheng Zhao[3], Xijun Liu [1,4✉], Yifan Liu [5], Rui Cao[6✉], Longchao Zhuo [7], Junqiang Ren[8], Jiaqiang Sun[9], Jun Luo [1] & Xuping Sun [10✉]

Electrochemical carbon monoxide reduction is a promising strategy for the production of value-added multicarbon compounds, albeit yielding diverse products with low selectivities and Faradaic efficiencies. Here, copper single atoms anchored to $Ti_3C_2T_x$ MXene nanosheets are firstly demonstrated as effective and robust catalysts for electrochemical carbon monoxide reduction, achieving an ultrahigh selectivity of 98% for the formation of multicarbon products. Particularly, it exhibits a high Faradaic efficiency of 71% towards ethylene at −0.7 V versus the reversible hydrogen electrode, superior to the previously reported copper-based catalysts. Besides, it shows a stable activity during the 68-h electrolysis. Theoretical simulations reveal that atomically dispersed Cu–O₃ sites favor the C–C coupling of carbon monoxide molecules to generate the key *CO-CHO species, and then induce the decreased free energy barrier of the potential-determining step, thus accounting for the high activity and selectivity of copper single atoms for carbon monoxide reduction.

[1] Institute for New Energy Materials & Low-Carbon Technologies and Tianjin Key Lab of Photoelectric Materials & Devices, School of Materials Science and Engineering, Tianjin University of Technology, Tianjin 300384, China. [2] School of Material Science and Engineering, University of Jinan, Jinan 250022, China. [3] Department of Environmental Engineering, University of Science and Technology Beijing, Beijing 100083, China. [4] Key Laboratory of Civil Aviation Thermal Hazards Prevention and Emergency Response, Civil Aviation University of China, Tianjin 300300, China. [5] College of Physics and Optoelectronic Engineering, Shenzhen University, Shenzhen 518060, China. [6] Stanford Synchrotron Radiation Lightsource, SLAC National Accelerator Laboratory, Menlo Park, CA 94025, United States. [7] School of Materials Science and Engineering, Xi'an University of Technology, Xi'an 710048 Shanxi, China. [8] State Key Laboratory of Advanced Processing and Recycling of Nonferrous Metals, Lanzhou University of Technology, Lanzhou 730050 Gansu, China. [9] State Key Laboratory of Coal Conversion, Institute of Coal Chemistry, Chinese Academy of Sciences, Taiyuan 030001 Shanxi, China. [10] Institute of Fundamental and Frontier Sciences, University of Electronic Science and Technology of China, Chengdu 610054 Sichuan, China. [11] These authors contributed equally: Haihong Bao, Yuan Qiu, Xianyun Peng, Jia-ao Wang. ✉email: xjliu@tjut.edu.cn; caorui@stanford.edu; xpsun@uestc.edu.cn

The electrochemical $CO_2$ reduction ($CO_2R$), as an appealing approach toward $CO_2$ mitigation and artificial carbon recycling, has been extensively studied in recent years[1–4]. However, effective strategies for the direct reduction of $CO_2$ to more value-added $C_{2+}$ products are not available at present due to the relatively low selectivity. In contrast, a wealth of efficient and selective electrocatalysts have been developed for the reduction of $CO_2$ to CO as the primary product, achieving a Faradaic efficiency (FE) larger than 90%[5–7]. In this regard, the further electrochemical CO reduction (COR) is desirable for deriving more value-added $C_{2+}$ products[6,8–10].

Abundant efforts have been made in developing efficient COR catalysts[11–13], however, to date, copper (Cu)-based materials are still the only known metal catalysts for electrochemically converting CO into multicarbon hydrocarbons and oxygenates, like $C_2H_4$ and $C_2H_5OH$ (EtOH), with an appreciable activity[14–17]. Generally, the stepped or kinked surfaces, as well as grain boundary surface terminations in polycrystalline Cu and oxide-derived Cu species[11,18,19], are considered to be active for COR. However, the structural complexity of these Cu species imposes a lot of difficulties to identify the exact sites responsible for their catalytic properties. Moreover, the diverse active centers of Cu species lead to unsatisfactory FEs and selectivities of the desired products because of various undetermined reaction pathways and competing hydrogen evolution reaction (HER). In contrast, single-atom (SA) catalysts exhibit unique active centers that can acquire an exceptional activity and selectivity in those electrocatalysis reactions involving multiple pathways[20–23]. Therefore, SA-based materials present potentially promising alternative catalysts, although they have never been scrutinized as COR electrocatalysts until now.

Recently, the two-dimensional (2D) materials, as a powerful platform to support SA catalysts, have attracted great attention because of large specific surface areas, more exposed active sites, and superior catalytic activities[24–30]. Particularly, with excellent electronic conductivity, catalytically active basal planes, and graphene-like unique layered structures, 2D $Ti_3C_2T_x$ MXene ($T_x$ represents surface functional groups) has been investigated extensively for a variety of electrochemical reactions[31–33]. More importantly, it has a unique feature of a high reducing capability, suitable surface defects, and hydrophilic surface functionalities, which make it an ideal candidate to support and stabilize SAs[34,35].

In this work, we disclose the synthesis, characterization, and COR activity of the Cu SA catalyst stabilized on 2D $Ti_3C_2T_x$ nanosheets. The supported Cu SA catalysts corresponded to O-coordinated Cu sites on $Ti_3C_2T_x$ matrix (Cu-SA/$Ti_3C_2T_x$), as revealed by X-ray absorption fine structure (XAFS) analysis. Computational studies confirmed the outstanding stability of Cu SAs on $Ti_3C_2T_x$. When applied in COR, Cu-SA/$Ti_3C_2T_x$ shows unprecedented selectivity (98% total) in the formation of $C_2$ products, which is much higher than its counterpart of Cu-NP/$Ti_3C_2T_x$. Outstandingly, the maximum FE of 71% towards $C_2H_4$ is achieved at −0.7 V versus the reversible hydrogen electrode (vs RHE), representing one of the highest values among the reported Cu-based COR catalysts[1,5,8–13]. Furthermore, Cu-SA/$Ti_3C_2T_x$ presents high electrochemical stability over 68 h. Theoretical analysis of pathways for $C_2H_4$ and EtOH formation gives an in-depth understanding of the enhanced reactivity and selectivity of Cu-SA/$Ti_3C_2T_x$ compared with Cu nanoparticles.

## Results

### Synthesis and characterization of Cu-SA/$Ti_3C_2T_x$. As illustrated in Fig. 1a, ultrathin $Ti_3C_2T_x$ nanosheets were prepared from parent $Ti_3AlC_2$ via etching in a mixed solution of HCl and LiF, and Cu-SA/$Ti_3C_2T_x$ was achieved by a one-step synthesis strategy

(see the Methods section for more details). The X-ray diffraction (XRD) pattern of Cu-SA/$Ti_3C_2T_x$, as shown in Fig. 1b, reveals a crystal structure similar to that of $Ti_3C_2T_x$ (ref. [33]). No diffraction peaks of any Cu species are observed, indicating the good dispersion of Cu SAs on $Ti_3C_2T_x$. High-angle annular dark-field scanning transmission electron microscopy (HAADF-STEM) and transmission electron microscopy (TEM) images clearly indicate a nanosheet morphology of the as-synthesized Cu-SA/$Ti_3C_2T_x$ (Supplementary Figs. 1, 2 and Fig. 1c). Cu-SA/$Ti_3C_2T_x$ exhibits some interlayer-stacked mesopores resulting from the nanosheet structure (Supplementary Fig. 3), which improves the accessibility of active sites, and thus, the overall catalytic performance of the catalyst[36]. No Cu particles are visible in the TEM images (Fig. 1c and Supplementary Fig. 2), which correlates well with the XRD results (Fig. 1b). Atomic-resolution HAADF-STEM images (Fig. 1d and Supplementary Fig. 4) taken from randomly selected regions show individual Cu atoms (sharp bright dots) uniformly dispersed on the $Ti_3C_2T_x$ crystal lattice fringes. The presence of Cu SAs can also be confirmed by the simulated HAADF-STEM images in Supplementary Fig. 5 and comparisons with the images of $Ti_3C_2T_x$ support in Supplementary Fig. 6. Besides, statistical analysis indicated that the average distance between each Cu atom is determined to be 0.61 nm (Supplementary Fig. 7), much higher than that of Cu−Cu bond (~0.27 nm) in Cu dimers. Even a small number of Cu pairs (<4%) are potentially formed, but such a small portion does very little contribution to the high COR activity. Energy-dispersive X-ray spectroscopy (EDX) further demonstrates the homogenous distribution of Ti, C, O, and Cu atoms over the Cu-SA/$Ti_3C_2T_x$ surface (Supplementary Fig. 8). A Cu loading of 0.2 wt% was determined by inductively coupled plasma optical emission spectrometry (ICP-OES). In addition, when more Cu precursor was added, the co-existence of Cu SAs and abundant Cu nanoclusters could be observed in the $Ti_3C_2T_x$ support (denoted as Cu-NC/$Ti_3C_2T_x$, Supplementary Fig. 9).

The chemical structure of Cu anchored to $Ti_3C_2T_x$ was further confirmed by X-ray photoelectron spectrometry (XPS) and X-ray absorption spectroscopy (XAS). The Cu 2$p$ XPS spectrum of Cu-SA/$Ti_3C_2T_x$ (Supplementary Fig. 10) reveals binding energies of 932.5 and 952.4 eV for Cu $2p_{1/2}$ and $2p_{3/2}$ orbitals, respectively, which are close to those of $Cu^0$ or $Cu^{1+}$ (refs. [37,38]). X-ray absorption near-edge structure (XANES) and extended X-ray absorption fine structure (EXAFS), which have been widely used for characterizing the SA catalysts in pioneering works[39,40], were performed at the Cu K-edge. The Cu foil, $Cu_2O$, and CuO were also tested as a comparison to Cu-SA/$Ti_3C_2T_x$. The XANES profiles (Fig. 1e) suggest that the Cu valence state in Cu-SA/$Ti_3C_2T_x$ is likely to be higher than that of metallic $Cu^0$ but lower than that of $Cu^{1+}$. The Fourier transform-EXAFS (FT-EXAFS) curve of Cu-SA/$Ti_3C_2T_x$ (Fig. 1f) shows one main peak at about 1.6 Å, and no prominent Cu−Cu back-scattering peak (2.2 Å, existed in Cu foil EXAFS spectra) is observed[23,41]. The wavelet transform (WT) contour plot of the Cu K-edge EXAFS oscillations of Cu-SA/$Ti_3C_2T_x$ (Supplementary Fig. 11) shows one intensity maximum at 4.0 Å$^{-1}$ associated with the Cu–O path in the first coordination shell. Compared with the WT plot of the Cu foil, no intensity maximum is detected near 7.7 Å$^{-1}$ (corresponding to the Cu–Cu path), this provides further evidence for the atomic dispersion of Cu atoms. According to the fitting results shown in Fig. 1g and Supplementary Fig. 12, the main peak at ~1.6 Å is associated with the nearest O coordination shell of the Cu-SA/$Ti_3C_2T_x$ catalyst (phase uncorrected). The small bulge at a radical distance of ~2.7 Å is associated with the scattering path of the second shell Ti, which is in agreement with the theoretical optimized model (inset of Fig. 1g). The fitted parameters are listed in Supplementary Table 1 and the coordination number of O is ~3. Therefore, Cu–$O_3$ sites formed

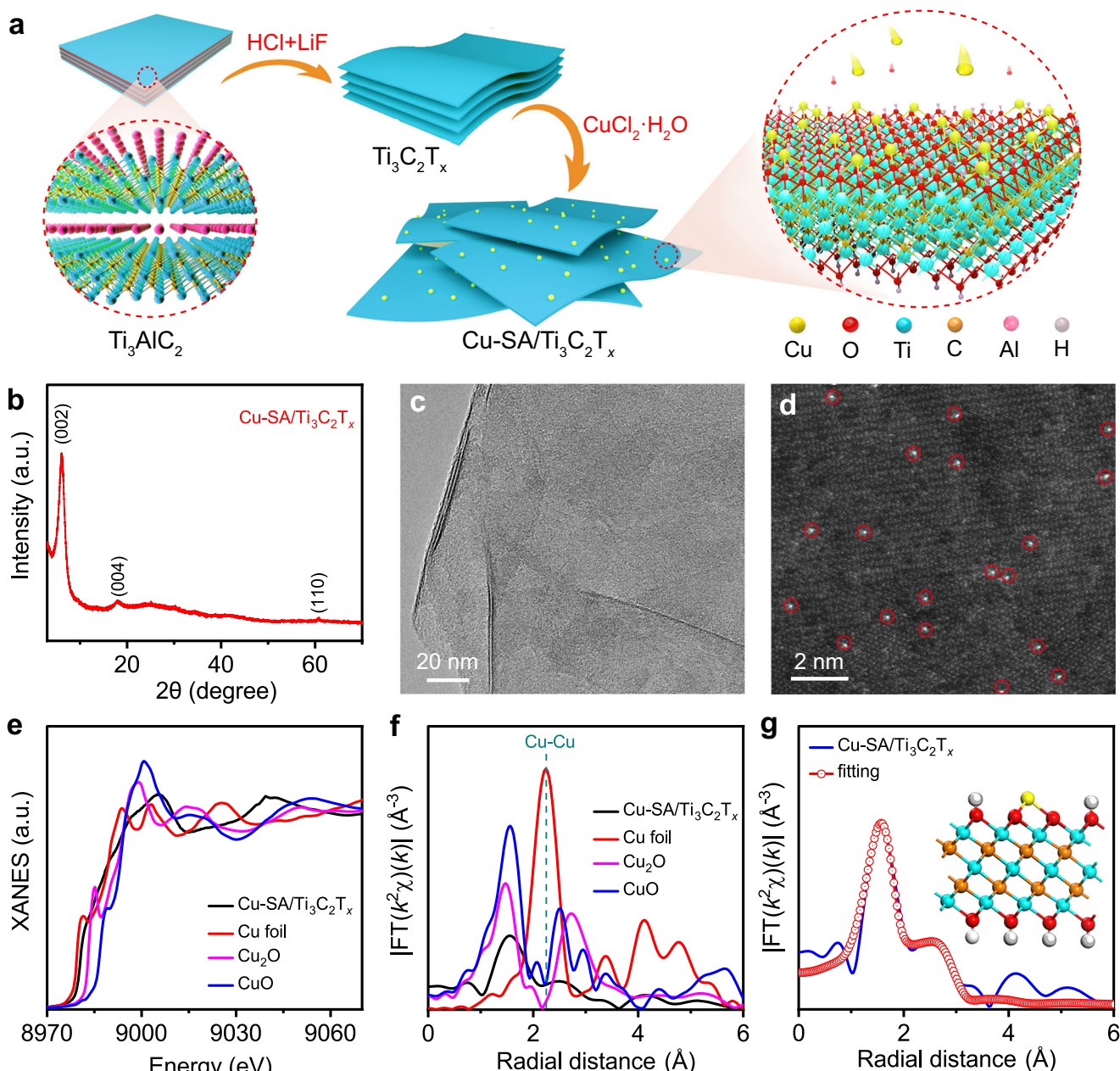

**Fig. 1 Preparation and structural characterization of Cu-SA/Ti₃C₂Tₓ. a** Schematic illustration of the synthesis procedure. **b** XRD pattern. **c** TEM image. **d** HAADF-STEM image in which some of the Cu SAs are highlighted by red circles. **e** XANES spectra at the Cu K-edge with CuO, Cu₂O and Cu foil as reference. **f** The $k^2$-weighted Fourier transform (FT) EXAFS curves in which $\chi(k)$ denotes the EXAFS oscillation function. **g** EXAFS fitting curve of Cu-SA/Ti₃C₂Tₓ, insert is an illustration of Cu-SA/Ti₃C₂Tₓ structure. The yellow, blue, dark yellow, red and white balls represent Cu, Ti, C, O and H, respectively.

in Cu-SA/Ti₃C₂Tₓ (inset of Fig. 1g). Moreover, the Bader charge analysis of the optimized model indicates that Cu SAs in Cu-SA/Ti₃C₂Tₓ are positively charged (+0.42, Supplementary Fig. 13), in line with XANES result (Fig. 1e). Unfortunately, the CO diffuse reflectance infrared Fourier transform spectroscopy (CO-DRIFTS) measurement under ambient conditions failed to probe the dispersion of Cu in Cu-SA/Ti₃C₂Tₓ (Supplementary Fig. 14), probably because of the weakening CO adsorption on the surface of Cu-SA/Ti₃C₂Tₓ without an applied bias voltage. Since the oxygen atoms are part of Tₓ surface groups, this suggests that the Cu atoms are dispersed on the surface of the Ti₃C₂Tₓ nanosheets.

**COR performance of Cu-SA/Ti₃C₂Tₓ in the alkaline system.** COR properties of Cu-SA/Ti₃C₂Tₓ were evaluated using a

Nafion-separated H-type cell with CO-saturated 1 M KOH as electrolyte. The potentials were converted to the RHE scale. Linear sweep voltammetry (LSV; Fig. 2a) shows that the current densities (normalized by the geometrical surface area) increase with increasing applied potential. The current density of Cu-SA/Ti₃C₂Tₓ in CO-purged electrolyte is distinctly higher than that under Ar atmosphere, demonstrating good electrocatalytic activity toward the COR. Electrolysis was performed in a potential range of –0.4 to –0.9 V vs RHE for 2 h at each constant potential. The gas and liquid products were quantified by gas chromatography (GC) and ¹H nuclear magnetic resonance (NMR) spectroscopy, respectively, in which dimethyl sulfoxide (DMSO) was used as an internal standard (Supplementary Fig. 15).

Figure 2b shows the FEs of the reduction products as well as the competing hydrogen formation on Cu-SA/Ti₃C₂Tₓ at the

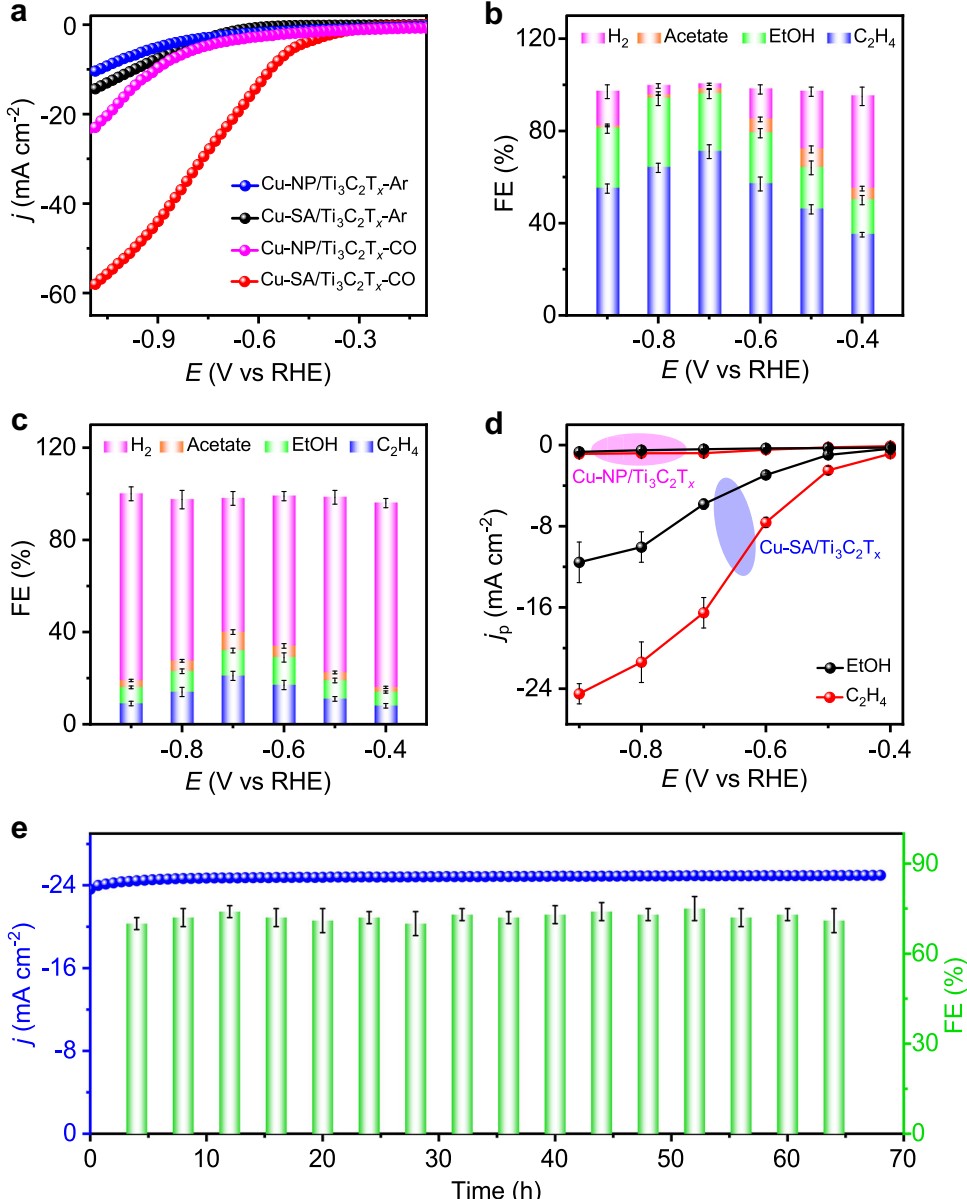

**Fig. 2 Electrochemical CO reduction performances. a** Linear sweep voltammetry curves of Cu-SA/Ti$_3$C$_2$T$_x$ measured in CO- or Ar-saturated 1 M KOH solution at a scan rate of 10 mV s$^{-1}$; potentials have been iR-corrected. **b, c** Faradaic efficiencies (FE) toward H$_2$ and CO reduction products (C$_2$H$_4$, EtOH, and acetate) of **b** Cu-SA/Ti$_3$C$_2$T$_x$ and **c** Cu-NP/Ti$_3$C$_2$T$_x$ at various applied potentials for 2 h. **d** The partial current densities ($j_p$) of C$_2$H$_4$ and EtOH at test potentials. **e** Chronoamperometry test and FE of C$_2$H$_4$ for Cu-SA/Ti$_3$C$_2$T$_x$ at an applied constant potential of −0.7 V vs RHE for 68 h. The error bars correspond to the standard deviations of measurements over three separately prepared samples under the same testing conditions.

applied potentials. A high total FE of 98% (−0.7 V vs RHE) for C$_2$ products has been achieved, suggesting the effective suppression of the competing HER. This value is comparable with those of Cu nanoflowers[42] (~100% FE for C$_{2+}$ products) and Cu nanoparticles[9] (~91% FE for C$_{2+}$ products). In the wide potential range from −0.6 to −0.9 V vs RHE, C$_2$ production reveals a total product selectivity of more than 79%, which declined at more negative potentials due to mass transport limitations of CO[5,43]. C$_2$H$_4$ is exclusively produced as the gas product of COR for Cu-SA/Ti$_3$C$_2$T$_x$, obtaining a maximum FE of 71% at −0.7 V vs RHE, which is significantly larger than the highest value of 52.7% reported for Cu NPs[44] (Supplementary Table 2). Meantime, in the liquid product, the FE of EtOH for Cu-SA/Ti$_3$C$_2$T$_x$ reached the highest value of 30% at −0.8 V vs RHE. At −0.7 V vs RHE, the FEs of EtOH and acetate for Cu-SA/Ti$_3$C$_2$T$_x$ were 25.0 and 2.2%,

respectively, and the corresponding formation rates are 2.79 and 0.44 mM h$^{-1}$, respectively (Supplementary Fig. 16). Furthermore, the partial current densities ($j_p$) of C$_2$H$_4$ and EtOH for Cu-SA/Ti$_3$C$_2$T$_x$ were as high as −16.5 and −5.8 mA cm$^{-2}$, respectively, at −0.7 V vs RHE (Fig. 2d).

Cu-SA/Ti$_3$C$_2$T$_x$ shows good stability over 68 h electrolysis after the initial activation of the catalyst layer in the first 30 min (Fig. 2e). An average current density of −24.8 mA cm$^{-2}$ is observed over the time of the long-term electrolysis. The corresponding FE for C$_2$H$_4$ is larger than 70% along with a ~3% oscillation caused by bubble accumulation and a sudden flush out[12]. HAADF-STEM images, as well as XRD and XPS results of Cu-SA/Ti$_3$C$_2$T$_x$ after electrochemical testing, indicate the well-preserved single Cu sites without the presence of any clusters and particles (Supplementary Figs. 17−19). Additional

density-functional-theory (DFT) calculations further confirmed the high kinetic stability of Cu SAs during the electrolysis (Supplementary Figs. 20 and 21).

To confirm the origins of carbon in the COR products, isotopic labeling experiments were performed by using $^{13}CO$ as the feeding gas. The $^{13}C$-labeled reduction products were analyzed by a quadrupole-type mass spectrometer (MS). Dominant peaks of $^{13}C$-$C_2H_4$ ($m/z = 30$), $^{13}C$-EtOH ($m/z = 47$), and $^{13}C$-acetic acid ($m/z = 62$) were observed (Supplementary Figs. 22–24)[45–47]. The evidence confirms that the evolved reduction products derive from the CO electrolysis over Cu-SA/$Ti_3C_2T_x$. In addition, the generated EtOH can also be excluded from the residual EtOH in preparing the working electrode (Supplementary Fig. 25).

As a comparison, Cu particles (10–50 nm) supported on $Ti_3C_2T_x$ (denoted as Cu-NP/$Ti_3C_2T_x$; Supplementary Figs. 26–30) were also prepared with a Cu loading of 5.2 wt%. Cu-NP/$Ti_3C_2T_x$ shows a very low current density of −16.2 mA cm$^{-2}$ at −1.0 V vs RHE (Fig. 2a), which is 3.2-fold lower than that on Cu-SA/$Ti_3C_2T_x$ (−52.2 mA cm$^{-2}$). The highest FEs of $C_2H_4$ and EtOH were only 21% (−0.7 V vs RHE) and 12% (−0.6 V vs RHE), respectively, in the electrocatalytic process on Cu-NP/$Ti_3C_2T_x$ (Fig. 2c), which are about 3.4- and 2.1-fold lower than on Cu-SA/$Ti_3C_2T_x$, respectively. In addition, the calculated formation rates of EtOH and acetate for Cu-NP/$Ti_3C_2T_x$ were 0.19 and 0.31 mM h$^{-1}$ at −0.7 V vs RHE, respectively (Supplementary Fig. 16). $H_2$ production dominated the whole potential range, showing an FE of 59–81% (Fig. 2c). The influence of Cu NP loading amount in Cu-NP/$Ti_3C_2T_x$ for the COR performance was also investigated. With an increase of Cu precursor in the synthetic process (Supplementary Figs. 30 and 31), ICP-OES results revealed that the Cu content in Cu-NP/$Ti_3C_2T_x$ increased from 5.2 to 9.8 and 20.3 wt%, respectively. We found that the two additional Cu-NP/$Ti_3C_2T_x$ control samples showed relatively improved COR activities (Supplementary Fig. 32); however, their performances are still inferior to those of Cu-SA/$Ti_3C_2T_x$ (Fig. 2a, b), suggesting that the Cu NP content is not the major contribution to the COR activity.

Meanwhile, the other two samples of pristine $Ti_3C_2T_x$ (Supplementary Fig. 6) and reduced $Ti_3C_2T_x$ (namely R-$Ti_3C_2T_x$, Supplementary Fig. 33) were prepared as comparation. As seen in Supplementary Figs. 34 and 35, the maximum FEs of $C_2H_4$ formation for $Ti_3C_2T_x$ and R-$Ti_3C_2T_x$ were 7.5% (−0.8 V vs RHE) and 5.2% (−0.7 V vs RHE), respectively, both of them are significantly lower than that of Cu-SA/$Ti_3C_2T_x$. This finding demonstrates that the $Ti_3C_2T_x$ support helps to capture and stabilize Cu species and the impact on the selectivity of COR is negligible. The high performance of Cu-SA/$Ti_3C_2T_x$ comes from the atomically dispersed Cu-related active sites.

To further confirm the activity of Cu SAs, the COR activity of Cu-SA/$Ti_3C_2T_x$ was examined in electrolyte containing 0.1 mM KSCN. SCN$^-$ anions are known to coordinate with Cu and poison the single Cu sites[48,49]. In the presence of SCN$^-$, Cu-SA/$Ti_3C_2T_x$ exhibits a noticeable decay in current density (Supplementary Fig. 36). Meanwhile, the obtained highest FE$_{C2}$ of Cu-SA/$Ti_3C_2T_x$ is as low as 26% (−0.7 V vs. RHE), much smaller than that of electrolyte without SCN$^-$ (Fig. 2b). These results strongly demonstrate that Cu single atoms act as the COR sites.

The increased electrochemical active surface area (ECSA) and interfacial charge transfer rate (Supplementary Figs. 37 and 38) may be potentially accounted for the enhanced COR activity of Cu-SA/$Ti_3C_2T_x$. As shown in Supplementary Fig. 39, the ECSA-corrected LSV results depict a much better catalytic current density of Cu-SA/$Ti_3C_2T_x$ in comparison with Cu-NP/$Ti_3C_2T_x$, implying that the presence of Cu SAs resulted in higher intrinsic activity. According to a previous study[42], the inferior COR

performance of Cu-NP/$Ti_3C_2T_x$ might result from the smaller ECSA in comparison with Cu-SA/$Ti_3C_2T_x$, which make it favor the competitive hydrogen evolution.

**Theoretical insights on COR activity**. To elucidate the COR activity, the mechanistic regimes for the formation of $C_2H_4$ and EtOH on Cu-SA/$Ti_3C_2T_x$ catalyst was parsed by calculating key intermediates and favorable reaction pathways in each primitive reaction, as shown in Fig. 3a, c and Supplementary Fig. 40. Generally, the $Ti_3C_2T_x$ surface is functionalized with $T_x$ groups, including −O, −OH, and −F. As revealed by the electron energy loss spectroscopy (EELS) spectrum in Supplementary Fig. 41, the residual F content in Cu-SA/$Ti_3C_2T_x$ is very low in comparison with O. Moreover, the −F is thermodynamically unfavorable compared with −O and −OH moieties[50]. In addition, in an aqueous solution, the −O termination would become hydroxylated under the electrocatalysis according to the surface pourbaix diagrams[51]. Therefore, in this work, the −OH terminated Cu-SA/$Ti_3C_2T_x$ model was employed, and the solvent effect is also considered using the implicit solvation model[52]. The primitive reaction on Cu (111) surface (Supplementary Fig. 27), which represents the predominate exposed active surface of Cu-NP/$Ti_3C_2T_x$, was also computed as a comparison.

The overall total reaction processes of $C_2H_4$ and EtOH are indicated below in equations (1) and (2), both of them are formed in 8e$^-$ reductions with $H_2O$ as the H$^+$ source:

$$2CO + 6H_2O + 8e^- \rightarrow C_2H_4 + 8OH^- \qquad (1)$$

$$2CO + 7H_2O + 8e^- \rightarrow C_2H_5OH + 8OH^- \qquad (2)$$

Since that the $2*CO \rightarrow *COCO$ process failed to proceed on Cu-SA/$Ti_3C_2T_x$ due to the large Cu-Cu distance in the catalyst (Supplementary Fig. 42), the *CHO pathway is considered for Cu-SA/$Ti_3C_2T_x$ (Fig. 3b). Firstly, one CO molecule adsorbed on the Cu SA site, resulting in the formation of *CO species (* denotes a binding site), and then *CHO formed by one proton transfer process. Next, *CHO was coupled with another CO by $*CHO + CO \rightarrow *CHO\text{-}CO$ (refs. [53,54]) with a free energy barrier of 0.32 eV, which is the rate-limiting step (RLS) during the whole COR process. For Cu (111), two C–C coupling pathways were calculated. *CO can be first hydrogenated into *COH with a free energy barrier of 1.18 eV (Supplementary Fig. 43), or two *CO coupled directly to from *COCO with a free energy barrier of 0.94 eV (Fig. 3b and Supplementary Fig. 44)[55,56]. The large free energy for these two C–C coupling step implies high-energy barriers for the CO-to-$C_2$ process, consistent with experimentally observed COR performance (Fig. 2c). To further verify the activation energy of the RLS, the climbing image nudged elastic band (CI-NEB) method was employed to calculate the transition states. As shown in Supplementary Fig. 45, the activation barrier for *CHO-CO formation on Cu-SA/$Ti_3C_2T_x$ is as low as 0.82 eV, which is much lower than that of the C–C coupling step on Cu (111) surface (1.36 eV). Therefore, this lower energy demand in the RLS of Cu-SA/$Ti_3C_2T_x$ accounts for its outstanding COR performance. In addition, the COR pathway was also calculated on pure $Ti_3C_2T_x$ as shown in Supplementary Fig. 46. It indicated that a large energy barrier of 1.29 eV is required for the activation of CO on $Ti_3C_2T_x$, much higher than that on Cu-SA/$Ti_3C_2T_x$. This agrees well with the poor COR performance of $Ti_3C_2T_x$ as discussed above.

The production selectivity of the COR reaction of Cu-SA/$Ti_3C_2T_x$ was then discussed. Overall, the reaction pathways to $C_2H_4$ and EtOH were identical in their beginning 4e$^-$ transfer process[57,58], which initially resulted in the formation of the *C-CHOH. After that, the proton-electron transfer of *C-CHOH

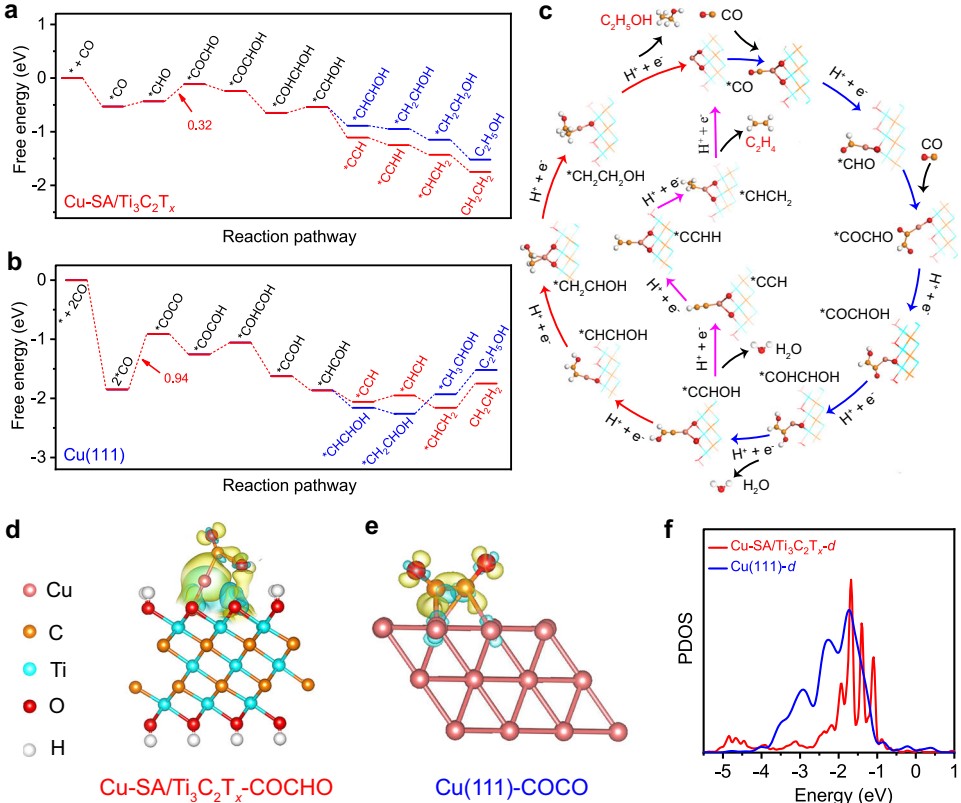

**Fig. 3 The optimized energy pathway for CO reduction toward C₂H₄ and EtOH on Cu-SA/Ti₃C₂Tₓ. a, b** Free energy diagrams of CO reduction over **a** Cu-SA/Ti₃C₂Tₓ and **b** Cu (111). The reason for choosing Cu (111) is that Supplementary Fig. 27 indicates the preferential orientation of the Cu nanoparticles in Cu-NP/Ti₃C₂Tₓ is <111>. **c** Schematic representation of CO reduction reaction pathways. In some adsorption configurations, Cu–O bond was elongated to get the lowest energy state. **d, e** Charge density difference of the *COCHO-adsorbed and *COCO-adsorbed configuration in **d** Cu-SA/Ti₃C₂Tₓ and **e** Cu (111), respectively. **f** The projected densities of states (PDOS) of *d*-orbitals in Cu-SA/Ti₃C₂Tₓ and Cu (111) with an aligned Fermi level. Yellow and blue shadows represent charge accumulation and depletion in the space, respectively; the pinkish-orange, blue, dark yellow, red, and white spheres represent Cu, Ti, C, O, and H, respectively.

splits the pathway to C₂H₄ from the pathway to EtOH. The free energy drop of the *C-CHOH → *C-CH (−0.62 eV) toward the formation of C₂H₄ is greater than that of the analog and competitive step *C-CHOH → *CH-CHOH (−0.45 eV) in the EtOH formation pathway, suggesting a higher selectivity toward C₂H₄ formation. This may be caused by the greater C–C π bond unsaturation of *C-CH compared with *CH-CHOH, which leads to a more stable absorption on the surface of catalysts[56,59]. Similar results can also be observed in CO₂R process, generally, the production of C₂H₄ was higher than EtOH (refs. [59–61]).

Moreover, charge accumulation and depletion between C and Cu atoms are observed on the charge density difference maps of both Cu-SA/Ti₃C₂Tₓ and Cu (111) surfaces (Fig. 3d, e), which is an indicator for successful adsorption of key intermediates on the catalyst surfaces. To further elucidate the electronic structures, we compared the projected densities of states (PDOS) of the *d*-orbitals of Cu-SA/Ti₃C₂Tₓ and Cu (111). It is found that the peak of the *d*-orbital of Cu-SA/Ti₃C₂Tₓ is narrower than that of Cu (111) and shifts toward the Fermi level, demonstrating promoted electron transport, and thus, higher reactivity (Fig. 3f)[62,63]. In addition, the ideal selectivity of Cu-SA/Ti₃C₂Tₓ toward C₂H₄ and EtOH formation can also be attributed to the uniform catalytic sites resulting from the certain and simple structure, suppressing the formation of key intermediates toward diverse products (Supplementary Fig. 40). In contrast, the high selectivity nature for conventional catalysts is typically difficult to possess due to their diverse and complicated active sites. These theoretical calculation

results provide some mechanistic explanations for the good activity of Cu-SA/Ti₃C₂Tₓ.

Additional DFT calculations were performed to determine the free energy (ΔG_{H*}) of H adsorption on the catalyst surface, which has been identified as a descriptor of HER, as shown in Supplementary Fig. 47. It shows that the ΔG_{H*} of Cu-SA/Ti₃C₂Tₓ is −0.35 eV, much more negative than that on the Cu-NP/Ti₃C₂Tₓ (+0.18 eV). Accordingly, the proceeding of the HER process on the Cu-SA/Ti₃C₂Tₓ surface is more difficult than on the Cu-NP/Ti₃C₂Tₓ, agrees well with the experimental results in Fig. 2b, c.

In summary, Cu SAs were successfully anchored to Ti₃C₂Tₓ nanosheets (Cu-SA/Ti₃C₂Tₓ) and firstly used as catalysts for the COR. Cu-SA/Ti₃C₂Tₓ exhibits an ultrahigh selectivity of 98% for the formation of C₂ products and unprecedented selectivity for C₂H₄ production (71%). Experiment and theoretical calculations revealed that the O-coordinated Cu SAs are stable during the electrolysis. For COR, it can promote the formation of the key *CO-CHO intermediate, and collectively decrease the free energy barrier of the rate-determining step. Overall, we speculate that the good selectivity of Cu-SA/Ti₃C₂Tₓ toward the formation of C₂H₄ and EtOH can be attributed to its good reactivity and structural simplicity. Our work sheds new light on the design of more advanced SA electrodes for efficient energy conversion. Furthermore, the electrosynthesis of C₂H₄ and EtOH enabled by the presented catalysis could provide a highly selective and energetically efficient route to value-added chemicals and fuels using abundant industrial CO as starting material.

## Methods

**Synthesis of $Ti_3C_2T_x$.** Typically, 1 g of $Ti_3AlC_2$ powder was mixed with 1 g of LiF and 10 mL of HCl, and the obtained mixture was kept stirring for 24 h at 35 °C. The resulting solid residue was washed several times with deionized water and centrifuged at a speed of $1150 \times g$. Finally, $Ti_3C_2T_x$ was obtained by freeze-drying.

**Synthesis of $NaBH_4$ reduced $Ti_3C_2T_x$ ($R-Ti_3C_2T_x$).** For the synthesis of R-$Ti_3C_2T_x$, 50 mL of $Ti_3C_2T_x$ suspension (1 mg mL$^{-1}$) was mixed with 1 mL of $NaBH_4$ aqueous solution (10 mg mL$^{-1}$) and stirred for 30 min. Subsequently, the mixture was ultrasonicated for 1 h, and the R-$Ti_3C_2T_x$ was obtained by centrifugation and freeze-drying.

**Synthesis of $Cu-SA/Ti_3C_2T_x$ and $Cu-NC/Ti_3C_2T_x$.** In a typical synthesis procedure of $Cu-SA/Ti_3C_2T_x$, 50 mg of $Ti_3C_2T_x$ was firstly dissolved in 50 mL of water in a round-bottom flask, after that, the mixture was sonicated for 30 min to obtain a uniform dispersion. Then, a certain amount of 1 mg mL$^{-1}$ $CuCl_2\cdot2H_2O$ (0.67 mL for $Cu-SA/Ti_3C_2T_x$ and 2.01 mL for $Cu-NC/Ti_3C_2T_x$) was added into 25 mL of $Ti_3C_2T_x$ suspension slowly, and electromagnetic stirred for 30 min. Subsequently, the mixture was ultrasonicated for 1 h. After centrifuged and washed with deionized water, the solid residue was vacuum freeze-dried to obtain $Cu-SA/Ti_3C_2T_x$.

**Synthesis of $Cu-NP/Ti_3C_2T_x$.** For the synthesis of $Cu-NP/Ti_3C_2T_x$ samples, 50 mg of $Ti_3C_2T_x$ was first dissolving in 50 mL of deionized water, and then a certain amount of 1 mg ml$^{-1}$ $CuCl_2\cdot2H_2O$ aqueous solution (6.7, 13.4 or 26.8 mL) was added into the above $Ti_3C_2T_x$ suspension slowly, followed by stirring for 30 min. Then, 1 mL of hydrazine hydrate ($N_2H_4\cdot H_2O$) was added into the above resultant mixture quickly, with vigorous stirring for 2 h at room temperature. Subsequently, the resulting solution was centrifuged, and the obtained residue was washed by deionized water. Finally, $Cu-NP/Ti_3C_2T_x$-x was obtained by freeze-drying. The Cu content in the three samples was determined by ICP-OES measurements as 5.2, 9.8, and 20.3 wt%, respectively.

**Materials characterizations.** TEM and EDX mappings were performed using FEI Talos F200X instruments. Atomic-resolution HAADF-STEM images and EELS were taken using a Titan Themis 60-300 STEM equipment equipped with a spherical probe aberration corrector. XRD was performed on an X-ray diffractometer (Rigaku SmartLab 9 kW) with Cu Kα radiation ($\lambda = 0.154598$ nm) at a scan rate of 10° min$^{-1}$ from 3° to 70°. XPS characterizations were conducted by Thermo Scientific Al Kα XPS system (ESCALAB250Xi), and the binding energies were calibrated by setting the measured binding energy of C $1s$ to 284.8 eV. ICP-OES analysis was conducted on a Thermo iCAP RQ instrument. The elemental analysis was performed by an elemental analyzer (Vario EL cube). CO-DRIFTS measurements were carried out on a Thermo Scientific Nicolet 6700 Fourier transform infrared (FT-IR) spectrometer.

**XAFS measurements and analysis details.** The Cu K-edge X-ray absorption spectra were collected on the unfocussed 20-pole 2 T wiggler side-station beamline 7-3 at the Stanford Synchrotron Radiation Lightsource under standard ring conditions of 3 GeV and ~500 mA. The Si (220) double crystal monochromators were used for energy selection. The components from higher harmonics were diminished after detuning the monochromator by 30–40%. By using cellulose as a binder and diluent, the solid samples were pressed into pallets and installed on a cryogenic sample rod. During data collection, they were kept inside a liquid helium CryoIndustries cryocooler at ~10 K. A detector of Lytle or Canberra Germanium 30-element array was used, and all the EXAFS data were measured to $k = 15$ Å$^{-1}$ in fluorescence mode. Internal energy was calibrated by simultaneous measurement of the absorption edge of a Cu-foil standard sample which is placed between two ionization chambers situated after the sample probe. The SamView of the Sixpack software was used to process the XAS data obtained from the Germanium detector. The normalization of data was accomplished by subtracting the cubic spline using the Athena program, in which the edge jump was assigned to 1.0. The FT-EXAFS data fitting was completed by the Artemis software. The structural parameters of bond distance ($R$), coordination numbers ($N$), and the bond variance ($\sigma^2$, related to the Debye-Waller factor and the static disorders of the scattering atoms) were varied during the fitting process.

**Electrochemical COR.** Electrocatalytic properties of the catalysts were conducted using an electrochemistry workstation (CHI660D Shanghai Chenhua Instrument Co.) with an H-type electrochemical cell. The catholyte and anolyte compartments were separated using a Nafion-117 membrane to avoid possible diffusion. A graphite rod and a Hg/HgO electrode were used as the counter and reference electrodes, respectively. The electrolyte used for all COR experiments was 1 M KOH. A carbon paper loaded with samples was used as the working electrode. Typically, 6 mg of catalyst and 0.5 mL of Nafion solution (0.05 wt%) were dispersed into 0.5 mL of EtOH (or isopropanol) by sonication for 30 min. After that, the ink was dropped onto the carbon paper and dried naturally.

For the detection of the COR products, the electrolysis was carried out in fresh CO-saturated 1 M KOH aqueous solution for 2 h at each applied potential. During the electrolysis, CO was continuously delivered into the cathodic compartment at a constant rate of 20 sccm. Polarization curves were recorded by a scan rate of 10 mV s$^{-1}$. The long-term durability was examined by using a chronoamperometry method.

The cell outlet gas products were analyzed by a gas chromatograph (GC, 7890 Agilent) equipped with a PoraPLOT Q column and a molecular sieve column. A flame ionization detector (FID) and a thermal conductivity detector (TCD) were used to analyze the products.

The Faradaic efficiency of gas species ($FE_{gas}$) was calculated as follows:

$$FE_{gas}(\%) = \frac{N_k \times C_k \times V_{CO} \times t \times F \times 10^{-3}}{22.4Q} \tag{3}$$

where $N_k$ is the exchanged electron numbers to produce species $k$, $C_k$ is the concentration of the product, measured by GC; $V_{CO}$ is the CO flow rate; $t$ is electrolysis time; F is the Faradaic constant (96485 C mol$^{-1}$); $Q$ is the total charge amount.

The liquid products were analyzed using a $^1$H NMR spectrum tested on a NMR (Bruker AVANCE AV III 400) equipment with a sensitivity of 480: 1. Standard curves were first made using standard chemicals over the concentration range of interest (EtOH and acetate), with the internal standard DMSO in 1 M KOH[60]. The $^1$H NMR spectrum was measured with water suppression via a presaturation method. The linearity of the two standard curves is as high as 0.999 (Supplementary Fig. 15), indicative of good accuracy in determining the concentration of products. In this work, the detection limits of EtOH and acetate were 0.1 and 0.15 μg mL$^{-1}$, respectively. To quantify the liquid products, 0.5 mL fresh electrolyte electrolyzed at a determined potential for 2 h was mixed with 0.1 mL D$_2$O and 0.05 μL DMSO. The ratio of the peak areas of the obtained EtOH and acetate to the DMSO peak area were compared to standard curves to quantify the concentrations of the reaction products. Then, the values of the EtOH or acetate yield rate can be derived from the slopes of the curves made by plotting the EtOH or acetate concentrations vs reaction times.

The Faradaic efficiency of liquid products ($FE_{liquid}$) was calculated as follows:

$$FE_{Liquid}(\%) = \frac{n \times V \times F \times N_k}{Q} \tag{4}$$

where $n$ (mol) is the content of EtOH or acetate, based on the calibration of the NMR; $V$ is the electrolyte volume in the cathodic chamber; F is the Faradaic constant (=96485 C mol$^{-1}$); $N_k$ is the number of transferred to produce species $k$; $Q$ is the total charge amount at different applied potentials.

For the isotope-labeling experiment, the same COR electrolysis was performed except $^{13}$CO ($^{13}$C 99.99%, Sigma-Aldrich) was used as the feeding gas. The products containing C-isotope was determined by a gas chromatograph (GC) equipped with an Agilent 5977A mass selective (MS) detector. High-purity He (99.99%) was used as the carrier gas.

**Computational details.** All the DFT calculations were performed using the VASP package with VASPKIT code for post-processing the calculated data. Generalized gradient approximation (GGA) with the Perdew-Burke-Ernzerhof (PBE) functional is employed to treat the exchange-correlation energy. The interaction between core and valence electrons was described by the projected augmented wave (PAW) basis set. A converged cutoff was set to 500 eV. Only the electrons in brackets of Ti [$3d^24s^2$], Cu[$3d^{10}4s^1$], C[$2s^22p^2$], O[$2s^22p^4$], and H[$1s^1$] were treated as valence electrons. Implicit solvation corrections were applied, and the electrolyte was incorporated using the Poisson−Boltzmann model implemented in VASPsol[52], in which the relative permittivity of the media $\epsilon_r$ was chosen as 78.4. Zero damping DFT-D3 method was used to investigate weak intermolecular interactions. In geometry optimizations, the force convergence standard was set below 0.02 eV Å$^{-1}$. The $3 \times 3 \times 1$ Monkhorst−Pack $k$-point mesh was used for each $Cu-SA/Ti_3C_2T_x$ and Cu (111) slab optimization. To obtain electronic energy in the ground state accurately, $5 \times 5 \times 1$ grid was used to produce a self-consistent field. The bottom two layers were fixed to implement the free energy calculation of each intermediate. A 15 Å vacuum layer was constructed along the z-axis for each model. Contributions of zero-point energies (ZPE), enthalpy, entropy, and pH to the free energies were considered and calculated. The calculation details of the Gibbs free energy change (ΔG) were expressed as follows:

$$\Delta G = \Delta E + \Delta E_{ZPE} + \int C_p dT - T\Delta S + \Delta G_{pH} \tag{5}$$

where $\Delta E$ is the electronic energy difference between the free-standing and the adsorption states of the intermediates; $\int C_p dT$ is the enthalpic temperature correction (see details in Supplementary Tables 3 and 4). $\Delta E_{ZPE}$ and $\Delta S$ are the corrections of zero-point energy and variation of entropy, respectively (see details in Supplementary Tables 3 and 4). Frequencies <50 cm$^{-1}$ were set to 50 cm$^{-1}$. $\Delta G_{pH}$ is the free energy correction of pH and is calculated according to the equation below:

$$\Delta G_{pH} = k_B T \times pH \times \ln 10 \tag{6}$$

where $k_B$ is the Boltzmann constant. pH is set to 14 in this work.

The binding energy of Cu SA ($E_{Cu-SA/Ti_3C_2T_x}^{bind}$) and the cohesive energy of Cu bulk ($E_{Cu_{bulk}}^{coh}$) were calculated by the following equations:

$$E_{Cu-SA/Ti_3C_2T_x}^{bind} = -\left(E_{Cu-SA/Ti_3C_2T_x} - E_{Ti_3C_2T_x} - E_{Cu-SA}\right) \quad (7)$$

$$E_{Cu_{bulk}}^{coh} = -\frac{E_{Cu_{bulk}} - nE_{Cu-SA}}{n} \quad (8)$$

where $E_{Cu-SA}$, $E_{Cu_{bulk}}$, $E_{Ti_3C_2T_x}$ and $E_{Cu-SA/Ti_3C_2T_x}$ are the energies of a free single Cu atom, the Cu foil, the $Ti_3C_2T_x$ substrate slab, and Cu-SA/$Ti_3C_2T_x$ model, respectively; $n$ is the sum of Cu atoms in a Cu conventional cell.

To measure the difficulty of Cu atom migration, the energy barrier ($E_b$) was calculated according to the following equation:

$$E_b = E_{TS} - E_{IS} \quad (9)$$

where $E_{TS}$ is the total energy of transition state configuration; $E_{IS}$ is the total energy of the initial configuration.

The Bader charge analysis was performed to quantitatively estimate the charge state of the Cu atoms in Cu-SA/$Ti_3C_2T_x$ and $Cu_2O$.

Transition states of interest were searched by CI-NEB method with an electron step convergence criterion of $10^{-8}$ eV.

## Data availability

The data that support the findings of this study are available within the article (and its Supplementary Information files) and from the corresponding authors upon reasonable request.

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

## Acknowledgements

This work was financially supported by the National Key R&D Program of China (2017YFA0700104), the National Natural Science Foundation of China (22075211, 51971157, 51808037, 21601136, and 51761165012), the Tianjin Science Fund for Distinguished Young Scholars (19JCJQJC61800), and the Science and Technology Development Fund of Tianjin Education Commission for Higher Education (No. 2018KJ126). The authors acknowledge Beijing PARATERA Tech CO., Ltd. for providing HPC resources that have contributed to the research results reported within this paper. The authors also thank Stanford Synchrotron Radiation Lightsource (SSRL) BL7-3 for providing the beam time. R.C. acknowledges support from DOE funded LDRD program and SSRL.

## Author contributions

X.J.L. and X.P.S. proposed and designed the electrocatalysts. H.H.B., Y.Q., and X.Y.P. co-synthesized the catalyst samples and carried out the electrochemical measurements, to which Y.Y.M. and J.Q.S. assisted. Y.Q., J.A.W., and S.Z.Z. conducted the DFT-calculations. R.C. performed the XAS experiments. H.H.B., X.Y.P., and Y.F.L. analyzed the XAS results. J.Q.R. and L.C.Z. assisted in analyzing the experimental data. X.J.L., J.L., and X.P.S. co-performed the experimental design and the mechanism analysis, and they co-supervised the whole project. All authors discussed the results.

## Competing interests

The authors declare no competing interests.
