## [Peer Review File · Nature Communications]

Reviewers' Comments:

Reviewer #1:

Remarks to the Author:

The authors claim Cu single atoms on Ti₃C₂T_x MCene nanosheets. They then report CORR with 71% FE to ethylene and 98% formation of C₂ products. From XAS they claim evidence of atomically dispersed Cu-O₃ sites. They state that DFT supports their picture.

Applied claims: There is no urgent need to move away from fully Cu catalysts. But if the authors truly had achieved a major advance in FE to desirable products, and understood why, this could be of interest.

Figure 2: It's a little surprising that Cu NP catalyst 2a have such vastly lower total current densities. Was the Cu NP loading incredibly low? That would explain all the hydrogen Figure 2c. Tom Jaramillo has documented near-100% FE to C₂⁺ in CORR based on Cu so something is very badly wrong in the controls.

And, as noted above, the CORR FE to ethylene, and the CORR FE to C₂⁺, are good but not better than prior reports.

One explanation for their findings: in their notionally CuSA catalysts, they loaded up fairly well with Cu (nanoparticles, Cu ~ polycrystalline metal, or whatever). In their NP controls, they didn't.

Claims of materials composition. It is hard to prove a true single-atom catalyst. Visually the Figure 1d HAADF-TEM image seems to point in the right direction, though I believe that it is hard to prove that the resolution is sufficient to distinguish single atoms of Cu from pairs, or from a patchy monolayer. Perhaps controls would help here? E.g.

- Negative control: HAADF-TEM of the same Ti₃C₂T_x support, but without any Cu

- Positive control: HAADF-TEM of the same catalyst, but with somewhat more Cu, where the XAS shows that the oxidation state of Cu is showing some Cu-Cu

Also the usual issue with TEM exists that one can pick and choose regions. Some kind of objective means of picking field of view, and some statistical sampling, might help be more persuasive here. Presumably a deep XAS expert will review this paper. I am skeptical of the evidence overall in this work for single Cu atom catalysts, but someone who does XAS for a living can provide expert review on this topic. A key question is whether XAS truly is capable of resolving this point i.e. can it distinguish materials that are mostly SA, mostly double-atom, or a blend of various coordination numbers of Cu.

Computational work. This paper needs at least one expert computational catalysis referee. Figure 3d looks to me to show that the Cu-SA is in fact not set up as truly single atoms? It seems to be a monolayer of Cu on the support.

In sum: There is no striking applied advance. The experimental support for the single atom assertion is not persuasive. The computational work would need careful review by a deep expert if the work were to receive further consideration.

Reviewer #2:

Remarks to the Author:

The authors performed an interesting study on the Isolated Cu single sites for high-performance electroreduction of CO to multicarbon products. After carefully go through the whole manuscript, some problems can be found as follows:

(1) in the second paragraph of introduction section, the authors mentioned that the single atom catalysts for electrochemical CO₂ reduction (CO₂R). The authors can make further discussion in depth on the single atom catalysts for CO₂R and many two dimensional materials used as single atom catalysts due to the large specific surface area and precisely defined active center. The

authors can add one sentence, two dimensional materials is a powerful platform to design the single atom catalysts for CO₂R. and cite the following references:

J. Am. Chem. Soc. 2019, 141, 3630–3640

J. Am. Chem. Soc. 2015, 137, 2757–2762

ACS Sustainable Chem. Eng. 2018, 6, 15494–15502.

J. Mater. Chem. A, 2019, 7, 3805–3814

J. Mater. Chem. A, 2019, 7, 11944–11952

RSC Adv., 2019, 9, 27710–27719

Energy Environ. Mater. 2019, 2, 193–200

(2) The authors should make a comparison of the catalytic activity of the current Cu-containing single-atom catalyst with the reported Cu-containing CO₂R catalysts, such as, Cu single crystals, Cu nanoclusters, etc in literature, and highlight the features of present Cu-containing single atom catalyst. This can make the present work more solid and interesting.

(3) what is the advantageous of Ti₃C₂T_x used as substrate for Cu single site catalyst compared with other substrates?

(4) in the reference section, there are some problems in references citations.

in ref.16, the authors should double check the journal name < ChemElectroChem> to make sure it is correct.

in ref.22, the authors should double check the journal name < Sustain. Energ. Fuels> to make sure it is correct.

in refs.30 and 33, < Angew. Chem. Int. Ed> should be < Angew. Chem. Int. Ed.>

in ref.32, the German version <Angew. Chem.> should be updated to the international version, consistent with other references.

in ref.35, < J. Phy. Chem. Lett.> should be < J. Phys. Chem. Lett.>

in ref.40, < Energ. Environ. Sci.> should be < Energy Environ. Sci.>

After the authors corrected all the above issues with a minor revision, this paper can be published after a further round review.

Reviewer #3:

Remarks to the Author:

This manuscript by Bao et al. reports a record-level selectivity of > 90 % for the formation of C₂ products (ethylene, ethanol, Acetate) In CO electrochemical reduction. The catalyst is described as Cu single atoms (Cu SAs) anchored to Ti₃C₂T_x MXene nanosheets. Although the C₂ production selectivity reported here is very high, the present study and the conclusions made about the active sites are not sufficiently supported by the data presented. The data analysis of several of the experimental methods discussed is also not rigorous enough. Therefore, the reviewer cannot recommend the acceptance of this work in Nature communications at current state. However, if all these comments are addressed, I would like to reconsider it. The detailed comments could be considered are given below:

(1) The claim that Cu species in Cu-SA/Ti₃C₂T_x are atomically dispersed is questionable. Firstly, EXAFS is an average result of bulk phase and there is a bulge at radical distance of ~2.5Å, which is possible from Cu-Cu interaction. Moreover, fitting parameters of Figure 1g should be provided. Without fitting parameters, the fitting is meaningless and unconvincing. In addition, dimers cannot be ruled out in HAADF-STEM of Supplementary Figure 5. The author just marked spots with larger distance, how about the ones with short distance? CO FT-IR or DRIFT is suggested to probe the dispersion state of Cu in Cu-SA/Ti₃C₂T_x.

(2) In Electrochemical COR performances test, the control experiments of pristine Ti₃C₂T_x and NaBH₄ reduced Ti₃C₂T_x without Cu species need to be conducted and compared. In addition,

considering both Ti₃C₂T_x and carbon paper contain carbon species, C¹⁸O Isotope labeling experiment must be conducted to confirm the products is from CO reduction other than from the reduction of carbon species in Ti₃C₂T_x and carbon paper. Moreover, the authors used ethanol to prepare the catalyst ink, thus lead the selectivity of ethanol questionable. Can the authors exclude that ethanol measured in solution after electrochemistry is not partly due to 'leaching' of residual ethanol on catalysts?

(3) Can the authors provide an example NMR spectrum to demonstrate and illustrate the analysis of the liquid? What was the concentration of ethanol and acetate in the liquid, and can the authors express the formation of ethanol and acetate in terms of a rate? What's the detection limits of NMR test? The authors did not give information about the reaction time of each potential, so the data of selectivity give here is unconvincing. The authors show error bars in Figure 2 but did not give explanation how they are obtained. The details of the COR tests and other characterizations also need improvement.

(4) The authors compared the XRD patterns of fresh and used Cu-SA/Ti₃C₂T_x (Figure 1b and Supplementary Figure 18) states that an amorphous broad peak around 24° were formed After the stability test, and was caused by the exfoliation of nanosheets resulting from the sonication process, as well as the intercalation of electrolyte ions during the electrolytic process. Thus, it is very likely the surface structure of the catalysts has changed and some carbon species in Ti₃C₂T_x is leached during COR test. The authors need to address this problem.

(5) In DFT calculations, as it is questionable to claim that Cu species in Cu-SA/Ti₃C₂T_x are atomically dispersed (see above comment 1), the model used here will be unconvincing. The case of Cu dimers model is suggested included in calculation for comparison.

Reviewer #4:

Remarks to the Author:

The authors report a combined theory/experimental study revealing a single atom copper catalyst supported on MXene with high CO reduction performance. The Faradaic efficiency and selectivity to overall C₂ products appears to compare favorably with the literature and to Cu nanoparticles on the same support. The paper is fairly well written and clear in its presentation. This is an important work showcasing a successful application of using single atom catalysts for CORR with good stability and activity. This manuscript should be publishable pending revisions.

Comments regarding the computational portion of the work:

The justification for the computational model used here is very sparse. A model system with a single Cu atom on a pristine MXene is used; however the surface may become hydroxylated or contain impurities such as Cl and F from the synthesis. This will strongly impact the chemical bonding and activity of the single atom. The authors should provide justification the pure O termination under the reaction conditions.

What is the binding strength of the single atom on the surface, relative to bulk Cu? What is the kinetic stability of this site, in terms of surface mobility or agglomeration from single atom to dimer? This should match experimental observations of stability.

What is the charge state of the Cu single atom in the model? How does this compare with a $0 < x < +1$ charge assignment from XPS? One way would be to compare computed charges with charges in Cu₂O as a reference, for example.

The higher activity of the single atom over the nanoparticles is explained by the difference in the rate-limiting formation of the "COCO^H" intermediate, though the difference in free energy is very small (0.73 vs 0.78 eV). This is not entirely convincing and the authors should supplement this with at least the calculation of the activation barriers for the formation of this COCO^H intermediate (and ideally some other important intermediates) from transition state search. This may tell more

information about the difference between the single atom and the bulk surface, if the barrier for C-C coupling is improved on the single atom (for either steric or electronic reasons). The barriers may also explain the difference in selectivity between C₂H₄ and EtOH.

Another experimental observation is the suppressed or decreased favorability of competing HER in the single atom versus the nanoparticle. Do the free energy of adsorption of hydrogen (GH) on the computational models follow the same experimental trends? For example, the single atom may overbind H relative to the surface and suppress HER. This is easy to test and would provide more support for the model.

Another consideration into the different performance between the single atom and nanoparticles could be due to solvation. Explicit solvation is commonly included in computational studies for CO₂RR or approximated with energetic corrections. Due to the different local geometry of the single atom site the solvation shell and stabilization energies will likely differ. This should at least be addressed in text or via calculations as a possible explanation.

For free energy calculations (G), were ZPE and entropy contributions (at reaction temperature) included? They should be, and if so, should be mentioned in the computational method.

Pg 13 Line 210 "mechanical" should be mechanistic

Responses to the Comments

Responses to the comments of Reviewer #1:

Comment:

The authors claim Cu single atoms on $\text{Ti}_3\text{C}_2\text{T}_x$ Mxene nanosheets. They then report CORR with 71% FE to ethylene and 98% formation of C_2 products. From XAS they claim evidence of atomically dispersed Cu-O₃ sites. They state that DFT supports their picture.

Applied claims: There is no urgent need to move away from fully Cu catalysts. But if the authors truly had achieved a major advance in FE to desirable products, and understood why, this could be of interest.

Response:

Thank you very much indeed for reviewing our manuscript. We greatly appreciate your inspiring, useful, helpful and constructive comments. Details of the corresponding answer and changes made are described below point by point. For your convenience, all changes have been highlighted in yellow in the revised main text and the revised supplementary information (SI) files.

1. Figure 2: It's a little surprising that Cu NP catalyst 2a have such vastly lower total current densities. Was the Cu NP loading incredibly low? That would explain all the hydrogen Figure 2c. Tom Jaramillo has documented near-100% FE to C₂+ in CORR based on Cu so something is very badly wrong in the controls.

Response:

Thanks for your valuable comments here. The Cu loading content in the control sample (that is, Cu nanoparticles supported on $\text{Ti}_3\text{C}_2\text{T}_x$ nanosheets (Cu-NP/ $\text{Ti}_3\text{C}_2\text{T}_x$)) was 5.2 wt% measured by inductively coupled plasma-optical emission spectrometer

analysis. According to your suggestions, we have also carefully synthesized two other Cu nanoparticles control samples (denoted as Cu-NP/Ti₃C₂T_x-x; x is the actual Cu loading in wt%, and it is 9.8 or 20.3 here) with a higher Cu NP content by adding more Cu precursor. The corresponding characterization results and synthesis details of Cu-NP/Ti₃C₂T_x-x have been added as Supplementary Figures 28 and 29 of the revised SI file and in Pages 16 and 17 of the revised main text file, respectively. We have also examined the CO reduction (COR) activity of Cu-NP/Ti₃C₂T_x-x under the same conditions as those of Cu single atoms (SAs) supported by Ti₃C₂T_x nanosheets (namely Cu-SA/Ti₃C₂T_x, Figure 2a,b). The corresponding electrochemical performance is added as Supplementary Figure 30 in Page 31 of the revised SI file. It shows that Cu-NP/Ti₃C₂T_x-9.8 and Cu-NP/Ti₃C₂T_x-20.3 afford improved reduction current densities of -21.97 and -34.0 mA cm⁻² at -1.0 V versus the reversible hydrogen electrode (vs RHE) in comparison with that of Cu-NP/Ti₃C₂T_x (-16.2 mA cm⁻²). Meantime, Cu-NP/Ti₃C₂T_x-9.8 and Cu-NP/Ti₃C₂T_x-20.3 exhibited the maximum Faradaic efficiency (FE) of 54.0% and 59.5% at -0.7 V vs RHE for C₂ products, respectively. However, the obtained two indicators are still inferior to those of Cu-SA/Ti₃C₂T_x (-52.2 mA cm⁻² vs 98%), suggesting that the Cu NP content is not the major contribution to the COR activity. This comparison further confirmed that Cu SAs in Cu-SA/Ti₃C₂T_x can effectively catalyze CO electrolysis, leading to its high COR activity. We reasoned that the less electrochemically accessible surface and lower intrinsic catalytic activity of active sites induced the inferior COR performance of Cu-NP/Ti₃C₂T_x, which was verified by measuring the electrochemically active surface area (ECSA) and ECSA-corrected COR polarization curves of Cu-NP/Ti₃C₂T_x and Cu-SA/Ti₃C₂T_x (Supplementary Figure 15 in our initially submitted version, it has been renumbered as Supplementary Figure 36 in Page 37 of the revised SI file because some new figures have been added). Furthermore, the theoretical calculations confirmed that the introduction of Cu SAs will induce a relatively low barrier energy for the potential-determining step (Figure 3a,b), which indicates the CO electrolysis is more likely to occur over Cu-SA/Ti₃C₂T_x.

Indeed, Tom Jaramillo et al. (*Nat. Catal.* 2019, 2, 702) reported Cu nanoflowers, which exhibited a maximum FE of near-100% for C₂₊ products at -0.23 V vs RHE. This work is a significant advancement for Cu-based COR electrocatalysts. Following it, our Cu-SA/Ti₃C₂T_x has shown an impressive COR performance, such as a nearly 100% FE for C₂ products at -0.70 V vs RHE and remarkable stability over 65 h. Moreover, we have read through not only the above Tom Jaramillo's paper but also all other papers about Cu-based materials, in all of which no Cu SAs have been reported for COR. In addition, comparison of Tom Jaramillo's work and our designed control sample (Cu-NP/Ti₃C₂T_x) reveals that the low roughness factor of our designed Cu-NP/Ti₃C₂T_x sample (*ca.* 10.3) resulted in an enhanced hydrogen evolution reaction (HER) activity, and thus leading to an inferior activity for the formation of C₂ products.

In addition to Supplementary Figures 28–30, the description of the above content has been added in Page 11 of the revised main text file, and Page 31 of the revised SI file. The paper (*Nat. Catal.* 2019, 2, 702) has been cited as ref. 42 in the reference list of the revised main text file.

2. And, as noted above, the CORR FE to ethylene, and the CORR FE to C₂₊, are good but not better than prior reports. One explanation for their findings: in their notionally Cu SA catalysts, they loaded up fairly well with Cu (nanoparticles, Cu~ polycrystalline metal, or whatever). In their NP controls, they didn't.

Response:

Thank you very much for your helpful comments. Indeed, our designed Cu-SA/Ti₃C₂T_x achieved a total C₂ FE of ~98%, which is comparable to that of the best-reported results (please see details in Supplementary Table 2), such as Cu nanoflowers (~100% FE for C₂₊ products, *Nat. Catal.* 2019, 2, 702) and Cu nanoparticles (91% FE for C₂₊ products, ref. 9 in our initially submitted version). Strikingly, the obtained maximum

FE for ethylene was 71%, which is significantly larger than the highest value (52.7%, ref. 32 in our initially submitted version, it has been renumbered as ref. 44 because some new references have been added in the revised main text file) reported for Cu-based COR catalysts. This enhanced COR activity for Cu-SA/Ti₃C₂T_x can be attributed to the abundant exposed active sites (Supplementary Figure 34), the high intrinsic activity (Supplementary Figure 36) and the decreased free energy barrier of the potential-determining step (Figure 3a,b). While for Cu-NP/Ti₃C₂T_x, as discussed above in comment #1, the inferior COR activity mainly resulted from the low roughness factor, which suppresses the intrinsic activity for the COR. This viewpoint is consistent with the experimental results (Figure 2c), which showed a maximum FE of 81% for H₂ production. Furthermore, this finding is also confirmed by Tom Jaramillo's work (*Nat. Catal.* 2019, 2, 702).

The description of the above content has been added in Page 9 of the revised main text file. The paper (*Nat. Catal.* 2019, 2, 702) has been cited as ref. 42 in the reference list of the revised main text file.

3. Claims of materials composition. It is hard to prove a true single-atom catalyst. Visually the Figure 1d HAADF-STEM image seems to point in the right direction, though I believe that it is hard to prove that the resolution is sufficient to distinguish single atoms of Cu from pairs, or from a patchy monolayer. Perhaps controls would help here? E.g.

- **Negative control: HAADF-STEM of the same Ti₃C₂T_x support, but without any Cu**
 - **Positive control: HAADF-STEM of the same catalyst, but with somewhat more Cu, where the XAS shows that the oxidation state of Cu is showing some Cu-Cu**
- Also the usual issue with TEM exists that one can pick and choose regions. Some kind of objective means of picking field of view, and some statistical sampling, might help be more persuasive here.**

Response:

Thanks for your valuable comments and suggestion. The atomic resolution high-angle annular dark-field scanning transmission electron microscopy (HAADF-STEM) imaging is acquired by an aberration-corrected electron microscopy with a sub-Å-resolution. In these HAADF-STEM images, the image intensity is approximately proportional to the square of the atomic number (Z). Therefore, the isolated heavier Cu SAs can be discerned in the $\text{Ti}_3\text{C}_2\text{T}_x$ support because of a different Z -contrast between Cu and Ti. In this regard, the aberration-corrected HAADF-STEM imaging is a powerful and widely-accepted technology to distinguish the presence of Cu SAs (refs. 23 and 30 in our initially submitted version, ref. 30 has been renumbered as ref. 41 because some new references have been added in the revised main text file). As you suggested, a higher-magnification HAADF-STEM image is added as Figure 1d in Page 6 of the revised main text file. More high-magnification HAADF-STEM images taken from randomly selected regions of the sample have also been added as Supplementary Figure 4 in Page 5 of the revised SI file. These additional images show that many individual Cu atoms (sharp bright dots) were monodispersed in the field of view. Meantime, the corresponding intensity profile of these sharp bright spots showed that the half-width of the intensity profile is about 1.2 Å, which is consistent with the Cu atomic radius (theoretical value: 1.4 Å). The low-magnification HAADF-STEM images of Cu-SA/ $\text{Ti}_3\text{C}_2\text{T}_x$ are also added in Supplementary Figure 1 in Page 2, in which no Cu clusters can be observed. Besides, HAADF-STEM images of the $\text{Ti}_3\text{C}_2\text{T}_x$ support are added as Supplementary Figure 5 in Page 6 of the revised SI file. Comparison of the atomic-resolution HAADF-STEM images (Supplementary Figure 4 and Figure 5b) of the two samples clearly demonstrated that the presence of atomically dispersed Cu SAs in Cu-SA/ $\text{Ti}_3\text{C}_2\text{T}_x$.

When more Cu precursor was added in the synthetic process, the co-existence of Cu SAs and abundant Cu nanoclusters could be observed in the $\text{Ti}_3\text{C}_2\text{T}_x$ support (namely Cu-NC/ $\text{Ti}_3\text{C}_2\text{T}_x$), which was confirmed by the newly added Supplementary Figure 8a,b in Page 9 of the revised SI file. Further, the corresponding X-ray absorption

spectroscopy (XAS) results showed a peak at 2.2 Å (Supplementary Figure 8e), which can be ascribed to the Cu–Cu binding. However, in the Cu-SA/Ti₃C₂T_x sample, we did not observe a peak at 2.2 Å (Figure 1f), implying the absence of Cu–Cu coordination.

Indeed, when viewed at the micro-level, the presence of Cu SAs can be clearly observed by these atomic-resolution HAADF-STEM images (Figure 1d and Supplementary Figures 4 and 6). Besides, we further performed a statistical analysis of the distance between Cu atoms based on the HAADF-STEM image, and the result is added as Supplementary Figure 6 in Page 7 of the revised SI file. Based on the DFT-optimized Cu dimers model (please see details in Supplementary Figure 19 in Page 20 of the revised SI file), the corresponding distance of the Cu–Cu bond is *ca.* 0.27 nm. By comparing the statistical result and geometrically optimized Cu dimers model, we can conclude that above 96% of the Cu atoms in Cu-SA/Ti₃C₂T_x are atomically anchored to the Ti₃C₂T_x support. Meantime, because of the large distance between the single Cu atoms (average: 0.61 nm), it is unlikely to form a patchy Cu monolayer in Cu-SA/Ti₃C₂T_x.

To further confirm the atomic dispersion of Cu on the Ti₃C₂T_x support, the XAS measurements provide deep insights on the macro-scale. Based on the Fourier transform extended X-ray absorption fine structure (FT-EXAFS) analysis (please see details in Figure 1f in Page 6 of the revised main text file), it clearly indicated that the Cu atoms are atomically dispersed; Cu-Cu paths at 2.2 Å are absent. All these results demonstrated that no Cu NPs can be found, and Cu SAs are uniformly dispersed on the Ti₃C₂T_x support in the Cu-SA/Ti₃C₂T_x sample.

In addition to Supplementary Figures 1, 4–6 and 8, the description of the above content has been added in Page 5 of the revised main text file and Pages 5–7 and 9 of the revised SI file.

4. Presumably a deep XAS expert will review this paper. I am skeptical of the evidence overall in this work for single Cu atom catalysts, but someone who does XAS for a living can provide expert review on this topic. A key question is whether XAS truly is capable of resolving this point i.e. can it distinguish materials that are mostly SA, mostly double-atom, or a blend of various coordination numbers of Cu.

Response:

Thank you for your valuable comments. Indeed, in our work, Dr. Cao Rui (one of the corresponding authors) is an XAS expert, working for Stanford Synchrotron Radiation Lightsource since 2018. He has published more than 10 XAS-related papers, such as *Appl. Catal. B* 2020, 268, 118747; *Small* 2020, 16, 1906735; *Energy Environ. Sci.* 2019, 12, 3508; *ACS Catal.* 2019, 9, 11743; *J. Am. Chem. Soc.* 2020, 142, 8431; *Angew. Chem.* 2019, 131, 11846; *Nat. Nanotech.* 2020, 1, 390. Up to now, the XAS technologies are considered to be relatively reliable for the determination of SA-based catalysts, as reported by refs. 21–23 in our initially submitted version. Meanwhile, HAADF-STEM imaging is also used to give a direct observation of the distribution of single atoms. Compared with HAADF-STEM images which reflected regional information, XAS technologies give average information on the atomic structure and electronic structure, including the distance between adjacent atoms, the number and type of coordinated atoms, and the oxidation state of atoms. The successful synthesis of SAs can be directly determined by analyzing the coordination information of metal atoms through the R space data in EXAFS. Of note, some pioneering works (*Science* 2016, 352, 797; *Science* 2019, 364, 1091) have also used these two methods for the determination of SA-based catalysts.

To confirm the structure of Cu atoms in Cu-SA/Ti₃C₂T_x, we performed a fitting for the R region of FT-EXAFS spectra, and the fitting parameters were added in Supplementary Table 1 in Page 43 of the revised SI file. According to the fitting results, the main peak at ~1.6 Å is associated with Cu–O single scattering path which represents the first coordination shell of the Cu-SA catalyst (phase uncorrected). The

smaller peak at ~ 2.7 Å is associated with the average Cu–Ti single scattering path (second shell) in agreement with the theoretical optimized model. According to the previous reports (refs. 21–23 in our initially submitted version; *Science* 2016, 352, 797; *Science* 2019, 364, 1091), obvious metal–metal bonds can be distinguished in R region of FT-EXAFS data if the metal atom aggregates were formed. However, no Cu–Cu bonds (2.2 Å) can be observed in our work, indicative of the extremely small amount of metal dimers in Cu-SA/Ti₃C₂T_x. Furthermore, we made a statistical analysis of HAADF-STEM images as discussed in comment #3, which indicated that above 96% of the Cu atoms are atomically dispersed. Therefore, according to the XAS and HAADF-STEM results, it is concluded that almost all of the Cu sites in Cu-SA/Ti₃C₂T_x are single atoms.

As for the coordination numbers of Cu, it is extremely difficult and impractical to make a statistic analysis for the exact distribution of the coordination number of Cu SAs. Alternatively, from the perfect R space fitting results of Cu-SA/Ti₃C₂T_x model, it indicated an average coordination number of Cu–O of 3.2 ± 0.4 , which is also consistent with our Cu-SA/Ti₃C₂T_x models (see inset of Figure 1g). In fact, the XAF fittings method has been widely reported by previous reports in the determination of coordination numbers of single-atom catalysts (refs. 21–23 in our initially submitted version; *Science* 2016, 352, 797; *Science* 2019, 364, 1091).

In addition to Supplementary Table 1, the description of the above content has been added in Pages 5 and 7 of the revised main text file. The papers (*Science* 2016, 352, 797; *Science* 2019, 364, 1091) have been cited as refs. 39 and 40 in the reference list of the revised main text file.

5. Computational work. This paper needs at least one expert computational catalysis referee. Figure 3d looks to me to show that the Cu-SA is in fact not set up as truly single atoms? It seems to be a monolayer of Cu on the support.

Response:

Thank you for your helpful comments and suggestion. We are so sorry about the ambiguous illustration of Figure 3d, which may be caused by the absence of figure legends representing different atoms. In the revised main text file, a clearer model has been given; hopefully, it gives a clear description of Cu SAs.

As you suggested, a computational chemist has been invited to review our work, and the more detailed discussion can be found in the responses to the comments of Reviewer #4.

Responses to the comments of Reviewer #2:

Comment:

The authors performed an interesting study on the Isolated Cu single sites for high-performance electroreduction of CO to multicarbon products. After carefully go through the whole manuscript, some problems can be found as follows:

Response:

Thank you very much indeed for reviewing our manuscript. We greatly appreciate your inspiring, useful, helpful and constructive comments. Details of the corresponding answer and changes made are described below point by point. For your convenience, all changes have been highlighted in yellow in the revised main text and the revised supplementary information (SI) files.

1. in the second paragraph of introduction section, the authors mentioned that the single atom catalysts for electrochemical CO₂ reduction (CO₂R). The authors can make further discussion in depth on the single atom catalysts for CO₂R and many two dimensional materials used as single atom catalysts due to the large specific surface area and precisely defined active center. The authors can add one sentence, two dimensional materials is a powerful platform to design the single atom catalysts for CO₂R. and cite the following references:

J. Am. Chem. Soc. 2019, 141, 3630–3640

J. Am. Chem. Soc. 2015, 137, 2757–2762

ACS Sustainable Chem. Eng. 2018, 6, 15494–15502

J. Mater. Chem. A, 2019, 7, 3805–3814

J. Mater. Chem. A, 2019, 7, 11944–11952

RSC Adv., 2019, 9, 27710–27719

Energy Environ. Mater. 2019, 2, 193–200

Response:

Thanks for your constructive suggestion, according to which an in-depth discussion is added in Pages 3–4 of the revised main text file. Meantime, these papers (*J. Am. Chem. Soc.* 2019, 141, 3630; *J. Am. Chem. Soc.* 2015, 137, 2757; *ACS Sustainable Chem. Eng.* 2018, 6, 15494; *J. Mater. Chem. A* 2019, 7, 3805; *J. Mater. Chem. A* 2019, 7, 11944; *RSC Adv.* 2019, 9, 27710; *Energy Environ. Mater.* 2019, 2, 193) are very important for our work indeed and they have been cited as refs. 24–30 in the reference list of the revised main text file.

2. The authors should make a comparison of the catalytic activity of the current Cu-containing single-atom catalyst with the reported Cu-containing CO₂R catalysts, such as, Cu single crystals, Cu nanoclusters, etc in literature, and highlight the features of present Cu-containing single atom catalyst. This can make the present work more solid and interesting.

Response:

Thank you very much for pointing this out. According to your suggestion, the currently reported Cu-containing catalysts for COR have been added in Supplementary Table S2 in Pages 44–46 of the revised SI file. In our work, Cu-SA/Ti₃C₂T_x showed much better selectivities towards COR products compared with the reported Cu-containing COR catalysts. It achieved the total C₂₊ FE of ~98%, and an ethylene FE of 71% at –0.7 V vs. RHE, both of which are comparable to and outperform the best-reported results of Cu-based catalysts under similar conditions (please see details in Supplementary Table 2). Furthermore, it shows a robust activity during the 65-h electrolysis. Note that it is the first time that a SA catalyst is used for the COR electrolysis. In addition to Supplementary Table 2, the description of the above content has been added in Page 9 of the revised main text file.

3. What is the advantageous of $Ti_3C_2T_x$ used as substrate for Cu single site catalyst compared with other substrates?

Response:

Thank you for this helpful comment. Compared with other substrates, $Ti_3C_2T_x$ MXene shows great promise in electrochemical applications due to its excellent electronic conductivity, catalytically active basal planes, and graphene-like unique layered structures (refs. 24–26 in our initially submitted version, they have been renumbered as refs. 31–33 in the reference list because some new references have been added in the revised main text file). More importantly, it has unique features of a high reducing capability, suitable surface defects and hydrophilic surface functionalities. These unique properties of $Ti_3C_2T_x$ make it an ideal candidate to support and stabilize SAs (*J. Am. Chem. Soc.* 2019, 141, 4086; *J. Mater. Chem. A* 2020, 8, 8913). The above description has been added in Page 4 of the revised main text file. The papers (*J. Am. Chem. Soc.* 2019, 141, 4086; *J. Mater. Chem. A* 2020, 8, 8913) have been cited as refs. 34 and 35 in the reference list of the revised main text file.

4. in the reference section, there are some problems in references citations.

in ref.16, the authors should double check the journal name < ChemElectroChem> to make sure it is correct.

in ref.22, the authors should double check the journal name < Sustain. Energ. Fuels> to make sure it is correct.

in refs.30 and 33, <Angew. Chem. Int. Ed> should be < Angew. Chem. Int. Ed.>

in ref.32, the German version should be updated to the international version, consistent with other references.

in ref.35, <J. Phy. Chem. Lett.> should be < J. Phys. Chem. Lett.>

in ref.40, <Energ. Environ. Sci.> should be < Energy Environ. Sci.>

Response:

Thank you very much for pointing this out. We are so sorry for the mistakes we made. These typo errors have been corrected. Meantime, we have carefully checked and confirmed that no similar mistakes exist in the reference list of the revised main text file.

Responses to the comments of Reviewer #3:

Comment:

This manuscript by Bao et al. reports a record-level selectivity of > 90 % for the formation of C2 products (ethylene, ethanol, Acetate) In CO electrochemical reduction. The catalyst is described as Cu single atoms (Cu SAs) anchored to $Ti_3C_2T_x$ MXene nanosheets. Although the C2 production selectivity reported here is very high, the present study and the conclusions made about the active sites are not sufficiently supported by the data presented. The data analysis of several of the experimental methods discussed is also not rigorous enough. Therefore, the reviewer cannot recommend the acceptance of this work in Nature communications at current state. However, if all these comments are addressed, I would like to reconsider it. The detailed comments could be considered are given below:

Response:

Thank you very much indeed for reviewing our manuscript. We greatly appreciate your inspiring, useful, helpful and constructive comments. Details of the corresponding answer and changes made are described below point by point. For your convenience, all changes have been highlighted in yellow in the revised main text and the revised supplementary information (SI) files.

1. The claim that Cu species in Cu-SA/ $Ti_3C_2T_x$ are atomically dispersed is questionable. Firstly, EXAFS is an average result of bulk phase and there is a bulge at radical distance of ~ 2.5 Å, which is possible from Cu-Cu interaction. Moreover, fitting parameters of Figure 1g should be provided. Without fitting parameters, the fitting is meaningless and unconvincing. In addition, dimers cannot be ruled out in HAADF-STEM of Supplementary Figure 5. The author just marked spots with larger distance, how about the ones with short distance? CO FT-IR or DRIFT is suggested

to probe the dispersion state of Cu in Cu-SA/Ti₃C₂T_x.

Response:

Thank you for these constructive comments. We fully agree that the EXAFS reflects an average result of the bulk phase. However, up till now, the XAS technology is considered to be a reliable measurement for the determination of single-atom catalysts, as widely reported by the pioneering works (refs. 21, 23, 30 and 42 in our initially submitted version, refs. 30 and 42 have been renumbered as refs. 41 and 60 in the reference list because some new references have been added in the revised main text). In our initially submitted version, we have performed a fitting for the Fourier-transformed EXAFS spectra. According to the fitting results, the main peak at ~ 1.6 Å is associated with Cu–O single scattering path, which represents the first coordination shell of the Cu-SA catalyst (phase uncorrected). The bulge at a radical distance of ~ 2.7 Å is associated with the average of Cu–Ti single scattering path indicated by the fitting results using the theoretical optimized model (please see inset of Figure 1g). In comparison, the Cu–Cu reflection appeared at 2.2 Å for Cu foil, which is consistent with the previous reports (refs. 23 and 30 in our initially submitted version, ref. 30 has been renumbered as ref. 41 in the reference list because some new references have been added in the revised main text). As you suggested, the corresponding fitting parameters of Figure 1g have been given in Supplementary Figure 11 and Supplementary Table 1 in Page 12 and 43 of the revised SI file, respectively.

To further distinguish isolated Cu SAs in Cu-SA/Ti₃C₂T_x, we have performed statistical analysis of the distance between Cu atoms based on the obtained HAADF-STEM images, and the result is added as Supplementary Figure 6 in Page 7 of the revised SI file. Based on the DFT-optimized Cu dimers model (please see details in Supplementary Figure 19 in Page 20 of the revised SI file), the corresponding distance of the Cu–Cu bond is ca. 0.27 nm. By comparing the statistical result and geometrically optimized Cu dimers model, we can conclude that above 96% of the Cu

atoms in Cu-SA/Ti₃C₂T_x are atomically anchored to the Ti₃C₂T_x support. Only a small fraction of Cu pairs (less than 4%) are formed; however, it is important to keep in mind that such a small portion of Cu pairs does very little contribution to the high COR activity of Cu-SA/Ti₃C₂T_x.

In addition to the statistical results, according to the comments of Reviewer #1, the HAADF-STEM images of two control samples (i.e. Ti₃C₂T_x support and Cu nanoclusters on Ti₃C₂T_x (denoted as Cu-NC/Ti₃C₂T_x)), have also been added as Supplementary Figures 5 and 8 in Pages 6 and 9 of the revised SI file, respectively. By comparing the HAADF-STEM images of Cu-SA/Ti₃C₂T_x and Ti₃C₂T_x, we can also conclude that the Cu SAs uniformly anchored to the Ti₃C₂T_x support. Meanwhile, with the appearance of Cu aggregated sites in Cu-NC/Ti₃C₂T_x (new control sample), a prominent peak representing the Cu–Cu bond emerged in FT-EXAFS curves, which is added as Supplementary Figure 8e in Page 9 of the revised SI file. Accordingly, the XAS, HAADF-STEM images and the corresponding statistical results all confirmed the atomically dispersed Cu atoms in Cu-SA/Ti₃C₂T_x.

As you suggested, we also have performed the CO-DRIFTS measurements, and the corresponding results are added as Supplementary Figure 12 in Page 13 of the revised SI file. Unfortunately, there is no characteristic peak corresponds to linearly bonded CO on Cu SAs in Cu-SA/Ti₃C₂T_x. This finding may be ascribed to the weakening of CO adsorption on the surface of Cu-SA/Ti₃C₂T_x. It is widely accepted that CO-DRIFTS characterization is an effective method to confirm a few of noble metal SAs, such as Pt, Pd and Rh SAs (*Nat. Commun.* 2018, 9, 4454; *ACS Catal.* 2014, 4, 1546; *Nat. Commun.* 2020, 11, 954).

In addition to Table S1 and Supplementary Figures 5, 6, 8, 11 and 12, the description of the above content has been added in Pages 5 and 7 of the revised main text file, as well as Pages 6, 7, 9 and 13 of the revised SI file.

2. In Electrochemical COR performances test, the control experiments of pristine $\text{Ti}_3\text{C}_2\text{T}_x$ and NaBH_4 reduced $\text{Ti}_3\text{C}_2\text{T}_x$ without Cu species need to be conducted and compared. In addition, considering both $\text{Ti}_3\text{C}_2\text{T}_x$ and carbon paper contain carbon species, C^{18}O Isotope labeling experiment must be conducted to confirm the products is from CO reduction other than from the reduction of carbon species in $\text{Ti}_3\text{C}_2\text{T}_x$ and carbon paper. Moreover, the authors used ethanol to prepare the catalyst ink, thus lead the selectivity of ethanol questionable. Can the authors exclude that ethanol measured in solution after electrochemistry is not partly due to 'leaching' of residual ethanol on catalysts?

Response:

Thank you for your inspiring comments and helpful suggestion, according to which the COR performances of the pristine $\text{Ti}_3\text{C}_2\text{T}_x$ and NaBH_4 reduced $\text{Ti}_3\text{C}_2\text{T}_x$ (namely R- $\text{Ti}_3\text{C}_2\text{T}_x$) without Cu species have been prepared, their structural characterizations (including TEM, XRD and XPS) have been added as Supplementary Figures 5 and 31 in Page 6 and 32 of the revised SI file, respectively. Their COR activity has been examined and the results are added as Supplementary Figures 32 and 33 in Page 33 and 34 of the revised SI file, respectively. As seen, the two control samples (that is, $\text{Ti}_3\text{C}_2\text{T}_x$ and R- $\text{Ti}_3\text{C}_2\text{T}_x$) showed inferior catalytic performance compared to that of Cu-SA/ $\text{Ti}_3\text{C}_2\text{T}_x$. In detail, the maximum FEs of C_2H_4 formation for $\text{Ti}_3\text{C}_2\text{T}_x$ and R- $\text{Ti}_3\text{C}_2\text{T}_x$ were 7.5% (−0.8 V vs RHE) and 5.2 % (−0.7 V vs RHE), respectively, both of them are significantly lower than that of Cu-SA/ $\text{Ti}_3\text{C}_2\text{T}_x$ (71% at −0.7 V vs RHE). This finding demonstrates that the $\text{Ti}_3\text{C}_2\text{T}_x$ support help to capture and stabilize the Cu SAs and the impact on the selectivity of COR is negligible.

As you suggested, isotopic labeling experiments were performed to confirm that the carbon in the COR products was derived from CO. ^{13}CO was used as the feeding gas. The COR products were analyzed by a quadrupole-type mass spectrometer (MS). The obtained MS results are added as Supplementary Figures 20–22 in Pages 21–23 of

the revised SI file, and the experiment details have been added in Page 20 of the revised main text file. Mass fragments at $m/z = 27, 28, 29$ and 30 were observed, resulting from ^{13}C -ethylene (*ACS Cent. Sci.* 2017, 3, 853). Meantime, dominant peaks of $^{13}\text{CH}_3^{13}\text{CH}_2\text{OH}^+$ ($m/z = 47$) and $^{13}\text{CH}_3^{13}\text{COOH}^+$ ($m/z = 62$) were observed. Besides, other mass fragments from $^{13}\text{CH}_3^{13}\text{CH}_2\text{O}^+$ and $^{13}\text{CH}_3^{13}\text{COO}^+$ also produced during the MS measurements (*Angew. Chem. Int. Ed.* 2016, 55, 10650; *Nat. Catal.* 2019, 2, 86). Based on these results, the carbon source of the evolved gas and liquid reduction products are confirmed to be originated from the CO electrolysis.

Indeed, ethanol was used as a solvent in the preparation of the catalyst ink; however, it evaporated completely from the carbon paper in the thereafter drying process under an infrared lamp. Furthermore, in order to exclude the impact of possible residual ethanol on the COR electrolysis, a control experiment has been conducted to recheck the COR performance of Cu-SA/Ti₃C₂T_x and Cu-NP/Ti₃C₂T_x. In this control experiment, ethanol was replaced by isopropanol in the preparation of the ink. The corresponding COR results are added as Supplementary Figure 23a, b in Page 24 of the revised SI file, which shows that Cu-SA/Ti₃C₂T_x and Cu-NP/Ti₃C₂T_x achieved a FE for C₂ products of 98.5% and 41.5%, respectively, both of which are very close to those of FE values by using the ink containing ethanol. In addition, the maximum FE for ethanol production reaches 30% at -0.8 V vs. RHE when the ink-containing isopropanol was employed—equal to the obtained FE value in our initially submitted version. This additional control experiment demonstrated that the produced ethanol was not originated from the residual ethanol on the catalyst surface.

In addition, we have also performed the COR electrolysis at an open circuit for 2 h, and the corresponding ^1H nuclear magnetic resonance (NMR) spectrum is added as Supplementary Figure 23c in Page 24 of the revised SI file. No ethanol can be detected after the electrolysis. This further suggests that the ethanol was generated from the electroreduction of dissolved CO by Cu-SA/Ti₃C₂T_x.

In addition to Supplementary Figures 5, 20–23 and 31–33, the description of the above content has been added in Pages 10 and 11 of the revised main text file, and Page 24 of the revised SI file. The papers (*ACS Cent. Sci.* 2017, 3, 853; *Angew. Chem. Int. Ed.* 2016, 55, 10650; *Nat. Catal.* 2019, 2, 86) have been cited as refs. 45–47 in the reference list of the revised main text file.

3. Can the authors provide an example NMR spectrum to demonstrate and illustrate the analysis of the liquid? What was the concentration of ethanol and acetate in the liquid, and can the authors express the formation of ethanol and acetate in terms of a rate? What's the detection limits of NMR test? The authors did not give information about the reaction time of each potential, so the data of selectivity give here is unconvincing. The authors show error bars in Figure 2 but did not give explanation how they are obtained. The details of the COR tests and other characterizations also need improvement.

Response:

Thanks for your constructive comment and helpful suggestion. As you suggested, the details of the ^1H NMR measurements have been given in Pages 19 and 20 of the revised main text file. Specifically, the liquid products were analyzed using a ^1H NMR spectrum tested on a nuclear magnetic resonance (Bruker AVANCE AV III 400) equipment with a sensitivity of 480 : 1. Dimethyl sulfoxide (DMSO) was used as an internal standard for quantification. Standard curves were first made using standard chemicals over the concentration range of interest (ethanol and acetate), with the internal standard DMSO in 1 M KOH (ref. 40 in our initially submitted version, it has been renumbered as ref. 58 in the reference list because some new references have been added in the revised main text). The results are added as Supplementary Figure 13 in Page 14 of the revised SI file. The ^1H NMR spectrum was measured with water suppression via a presaturation method. The detection limits of ethanol and acetate were 0.1 and 0.15 $\mu\text{g mL}^{-1}$, respectively. In this work, the linearity of the two curves

was both better than 0.999, which demonstrated good accuracy in determining the concentration of products. 0.5 mL fresh electrolyte electrolyzed at a determined potential for 2 h was mixed with 0.1 mL D₂O and 0.03 μL DMSO. The ratio of the peak areas of the obtained ethanol and acetate to the DMSO peak area were compared to standard curves to quantify the concentrations of the reaction products. For example, after collected at -0.7 V vs RHE for 2 h, the concentrations of ethanol and acetate were measured to be 5.4 and 0.88 mM (in this work, 10 mL), respectively. Then, the yield rate of the ethanol or acetate can be derived from the slopes of the curves made by plotting the ethanol or acetate concentrations vs reaction times, which are added as Supplementary Figure 14 in Page 15 of the revised SI file. The formation rates of ethanol and acetate over the Cu-SA/Ti₃C₂T_x are calculated to be 2.79 and 0.44 mM h⁻¹, respectively. While for Cu-NP/Ti₃C₂T_x, the formation rates of ethanol and acetate are 0.19 and 0.31 mM h⁻¹, respectively.

Note that, the COR activities of the catalysts were evaluated using controlled potential electrolysis with CO-saturated electrolyte for 2 h.

In this work, the error bars correspond to the standard deviations of measurements over three separately prepared samples under the same testing conditions.

In addition to Supplementary Figures 13 and 14, the description of the above content and the details of the COR tests and other characterizations have been added in Pages 9, 11, and 17–20 of the revised main text file, as well as in Pages 14 and 15 of the revised SI file.

4. The authors compared the XRD patterns of fresh and used Cu-SA/Ti₃C₂T_x (Figure 1b and Supplementary Figure 18) states that an amorphous broad peak around 24° were formed after the stability test, and was caused by the exfoliation of nanosheets resulting from the sonication process, as well as the intercalation of

electrolyte ions during the electrolytic process. Thus, it is very likely the surface structure of the catalysts has changed and some carbon species in $\text{Ti}_3\text{C}_2\text{T}_x$ is leached during COR test. The authors need to address this problem.

Response:

Thank you for this helpful comment. According to the responses to your comment #2, the ^{13}C isotopic labeling experiment strongly confirmed that the carbon in the evolved products (including gas and liquid products) originates from the gaseous CO supplied. Further, the elemental analyzer (Vario EL Cube) was used to detect the content of carbon in Cu-SA/ $\text{Ti}_3\text{C}_2\text{T}_x$ before and after the long-term stability test. As expected, the carbon content of the fresh Cu-SA/ $\text{Ti}_3\text{C}_2\text{T}_x$ and the used one is measured to be 35.7 and 36.2 wt%, respectively, indicative of the good stability of Cu-SA/ $\text{Ti}_3\text{C}_2\text{T}_x$ during the COR test.

The description of the above content has been added in Page 17 of the revised SI file. In addition, the details of the element analysis have also been added in Page 17 of the revised main text file.

5. In DFT calculations, as it is questionable to claim that Cu species in Cu-SA/ $\text{Ti}_3\text{C}_2\text{T}_x$ are atomically dispersed (see above comment 1), the model used here will be unconvincing. The case of Cu dimers model is suggested included in calculation for comparison.

Response:

Thank you very much for your valuable comments. According to the responses to your comment #1 and the comments #3 and #4 from Reviewer #1 (please see details in the above responses), we can conclude that almost all of the single Cu atoms are isolatedly dispersed on the $\text{Ti}_3\text{C}_2\text{T}_x$ support. Therefore, the case of Cu dimers model is not taken into consideration in this work.

Responses to the comments of Reviewer #4:

Comment:

The authors report a combined theory/experimental study revealing a single atom copper catalyst supported on MXene with high CO reduction performance. The Faradaic efficiency and selectivity to overall C₂ products appears to compare favorably with the literature and to Cu nanoparticles on the same support. The paper is fairly well written and clear in its presentation. This is an important work showcasing a successful application of using single atom catalysts for CORR with good stability and activity. This manuscript should be publishable pending revisions.

Response:

Thank you very much indeed for reviewing our manuscript. We greatly appreciate your inspiring, useful, helpful and constructive comments. Details of the corresponding answer and changes made are described below point by point. For your convenience, all changes have been highlighted in yellow in the revised main text and the revised supplementary information (SI) files.

Comments regarding the computational portion of the work:

1. The justification for the computational model used here is very sparse. A model system with a single Cu atom on a pristine MXene is used; however the surface may become hydroxylated or contain impurities such as Cl and F from the synthesis. This will strongly impact the chemical bonding and activity of the single atom. The authors should provide justification the pure O termination under the reaction conditions.

Response:

Thank you for your valuable comments and suggestion. Indeed, the surface of

MXene is always functionalized with T_x groups, including $-O$, $-OH$ and $-F$. Besides, a slight amount of Cl element also can be detected (ca. 3 at % by XPS measurement in this work), however, most of the residual Cl can be washed off during the synthesis of $Ti_3C_2T_x$ MXene nanosheets (as described in the Methods). In addition, the composition of $Ti_3C_2T_x$ surface functional groups are influenced by the etching method of Ti_3AlC_2 . In this work, we used LiF and HCl as etching agent, therefore, the amount of $-O$ terminations is supposed to be higher than that of $-F$ and $-OH$ (*Phys. Chem. Chem. Phys.* 2016, 18, 5099). Moreover, EELS analysis also used to determine the elements present in the $Ti_3C_2T_x$ flakes, and the result has been added as Supplementary Figure 38 in Page 39 of the revised SI file. The EELS spectrum indicated signals of C K-edge, O K-edge, F K-edge, and Ti L-edge (*ACS Nano* 2016, 10, 9193). Clearly, the F K-edge at 685 eV is weaker than that of O K edge, suggesting the F content is very low in Cu-SA/ $Ti_3C_2T_x$. Meantime, the previous study confirmed the $-F$ is less thermodynamically in comparison with $-O$ groups (*Sci. Bull.* 2018, 63, 1397). Therefore, in our initially submitted version, we use the $-O$ terminated one as the model of Cu-SA/ $Ti_3C_2T_x$.

As you suggested, the $-O$ termination may become hydroxylated in an aqueous solution. Therefore, we carefully checked the thermodynamical stability of Cu-SA/ $Ti_3C_2T_x$ model under the reaction condition. According to the surface pourbaix diagrams of Ti_3C_2 (*ACS Catal.* 2017, 7, 494), the MXenes will be oxidized by H_2O and turn into $-OH$ terminated ones under a relatively negative applied potential. Therefore, in this revised version, we changed the O-terminated model into the OH terminated ones, and the relative DFT analysis has been recalculated based on this new model. The newly calculated results replaced the original version of Figure 3. In brief, the rate-limiting step (RLS) for Cu-SA/ $Ti_3C_2T_x$ was determined to be $*CHO + CO \rightarrow *CHO-CO$. In contrast, the RLS of Cu (111) was $2*CO \rightarrow *COCO$; however, the free energy barrier of the RLS is 0.94 eV, much higher than that of Cu-SA/ $Ti_3C_2T_x$ (0.32 eV). The theoretical calculation results agree well with the high COR activity of Cu-SAs from the experimental observation (Fig. 2b,c).

In addition to Supplementary Figure 38, the description of the above content has been added in Page 12 of the revised main text file. The papers (*ACS Nano* 2016, 10, 9193; *Sci. Bull.* 2018, 63, 1397; *ACS Catal.* 2017, 7, 494) have been cited as ref. S7, and refs. 48–49 in the reference list of the SI file and revised main text file, respectively.

2. What is the binding strength of the single atom on the surface, relative to bulk Cu? What is the kinetic stability of this site, in terms of surface mobility or agglomeration from single atom to dimer? This should match experimental observations of stability.

Response:

Thank you very much for proposing this vital comments. To estimate the stability of Cu single atom on $\text{Ti}_3\text{C}_2\text{T}_x$ support, the binding energy of Cu-SA/ $\text{Ti}_3\text{C}_2\text{T}_x$ ($E_{\text{Cu-SA}/\text{Ti}_3\text{C}_2\text{T}_x}^{\text{bind}}$) was determined. It is calculated to be +2.16 eV, smaller than the cohesive energy of +3.48 eV for bulk Cu ($E_{\text{Cu}_{\text{bulk}}}^{\text{coh}}$); however, the binding strength is still quite strong and agrees well with previous reports for SA catalysts (*Nanoscale* 2018, 10, 17893; *Nat. Commun.* 2016, 7, 1080). Besides, the kinetic stability that affecting the stability in practical electrocatalysis should be taken into consideration as well. Thus, to estimate the kinetic stability of Cu-SA/ $\text{Ti}_3\text{C}_2\text{T}_x$, the surface mobility of Cu SAs was discussed by simulating the mobilization of a single Cu atom on $\text{Ti}_3\text{C}_2\text{T}_x$ surface using the climbing image-nudged band (CI-NEB) approach. The optimized structures and their corresponding energy barrier profiles have been added as Supplementary Figure 18 in Page 19 of the revised SI file. All the possible migration paths were considered. It shows that the energy barrier for the mobilization of a single Cu atom from this most stable trap (1) to an adjacent triangle oxygen trap (1(7)) or Ti top site (12) is 1.17 eV and 1.23 eV, respectively. Another migration pathway is also

considered from the (1(7)) site to an adjacent oxygen top site (1(7)→11), which gives a larger energy barrier of 1.63 eV. This energy barrier is comparable to those reported for Pd SAs (1.67 eV) (*Chem. Mater.* 2017, 29, 9456) and Cu SAs (0.95 eV) (*Nanoscale* 2018, 10, 17893) on CeO₂ (111), which indicates that single Cu atom on the Ti₃C₂T_x surface is difficult to move away from the triangle oxygen traps, suggesting the high kinetic stability of Cu-SA/Ti₃C₂T_x.

Another possible instability formation for single Cu atoms is the appearance of Cu-dimer species, which are potential CO adsorption sites that would affect and change the total COR mechanism. Therefore, it is necessary to investigate, in the theoretical aspect, whether dimers can be formed in our highly dispersed Cu SAs during the electrolysis. The possible surface agglomeration from single Cu atom to Cu dimer paths and corresponding energy barriers have been calculated using CI-NEB method. The corresponding results have been added as Supplementary Figure 19 in Page 20 of the revised SI file. It illustrates that two adjacent Cu single atoms need a relatively high energy barrier of 1.23 eV to form a Cu dimer through atom migration, indicating that Cu dimer is difficult to be formed.

The good stability of Cu-SA/Ti₃C₂T_x can also be evidenced by the atomic-resolution HADDF-STEM images in Supplementary Figure 15, in which no nanoparticle or clusters were observed after the long-term stability test. Therefore, the consistency of the theoretical calculation and experimental results confirmed the good stability of Cu-SA/Ti₃C₂T_x in the electrocatalysis.

In addition to Supplementary Figures 18 and 19, the description of the above content has been added in Pages 19 and 20 of the revised SI file. The calculation details of the binding energy and diffusion barriers of the samples have also been added in Pages 20 and 21 of the revised main text file. The papers (*Nanoscale* 2018, 10, 17893; *Nat. Commun.* 2016, 7, 10801; *Chem. Mater.* 2017, 29, 9456) have been cited as refs. S4–S6 in the reference list of the revised SI file.

3. What is the charge state of the Cu single atom in the model? How does this compare with a $0 < x < +1$ charge assignment from XPS? One way would be to compare computed charges with charges in Cu_2O as a reference, for example.

Response:

Thank you very much for this helpful suggestion. Bader charge analysis has been performed to reveal the charge state of Cu single atom in $\text{Cu-SA/Ti}_3\text{C}_2\text{T}_x$, and the charge state of Cu in bulk Cu_2O was also calculated as a comparison. The results show that the charge state of the Cu in $\text{Cu-SA/Ti}_3\text{C}_2\text{T}_x$ and bulk Cu_2O are +0.420 and +0.525 respectively, which is consistent with the experimental results obtained by XANES result (Figure 1e). The description of the above content and the corresponding calculation details have been added in Pages 7 and 22 of the revised main text file, respectively.

4. The higher activity of the single atom over the nanoparticles is explained by the difference in the rate-limiting formation of the "COCOH" intermediate, though the difference in free energy is very small (0.73 vs 0.78 eV). This is not entirely convincing and the authors should supplement this with at least the calculation of the activation barriers for the formation of this COCOH intermediate (and ideally some other important intermediates) from transition state search. This may tell more information about the difference between the single atom and the bulk surface, if the barrier for C-C coupling is improved on the single atom (for either steric or electronic reasons). The barriers may also explain the difference in selectivity between C_2H_4 and EtOH.

Response:

We greatly appreciate your constructive and important comments. According to your

comment, we recalculated the free energy profile of to EtOH and C₂H₄ pathways on Cu-SA/Ti₃C₂T_x and Cu (111) using VASP package with VASPKIT code for post-processing of the calculated data. Generalized gradient approximation (GGA) with the Perdew-Burke-Ernzerhof (PBE) functional is used to treat the exchange-correlation energy. The projected augmented wave (PAW) basis set was used to describe the interactions between core and valence electrons. Zero damping DFT-D3 method was applied to study weak intermolecular interactions.

Besides, in this revised free energy calculation, the energy contribution of the solvent effect, free energy correction of pH, zero-point energy corrections, enthalpic temperature correction, and entropy contributions have been taken into consideration. The computation details have all been added in Pages 20–22 of the revised main text file. The new free energy profiles have replaced the original version of Figure 3a,b. As shown, the rate-limiting step of COR on Cu (111) became C–C coupling process after consideration of the above-mentioned corrections. Also, the transition state calculation has been performed on the “C–C coupling” step using CI-NEB method. The energy profiles, as well as the corresponding initial state (IS), transition state (TS) and final state structures (FS), have been added in Supplementary Figure 40 in Page 41 of the revised SI file. It shows that the activation energy barrier for CO dimerization on Cu-SA/Ti₃C₂T_x is as low as 0.82 eV, which is much lower than the energy barrier of 1.36 eV on Cu (111). This result reasonably explains the higher COR activity of the Cu-SA/Ti₃C₂T_x over the Cu-NP/Ti₃C₂T_x.

In addition to Supplementary Figure 40, the description of the above content has been added in Page 13 of the revised main text file.

5. Another experimental observation is the suppressed or decreased favorability of competing HER in the single atom versus the nanoparticle. Do the free energy of adsorption of hydrogen (GH) on the computational models follow the same

experimental trends? For example, the single atom may overbind H relative to the surface and suppress HER. This is easy to test and would provide more support for the model.

Response:

Thank you very much for this helpful suggestion, according to which we compared the reaction free energy profiles of the HER on Cu-SA/Ti₃C₂T_x and on the Cu (111) surface. The calculated free-energy diagrams have been added as Supplementary Figure 41 in Page 42 of the revised SI file. As seen, the difference of free energy (ΔG_{H^*}) of Cu-SA/Ti₃C₂T_x is -0.35 eV, much more negative than that on the Cu-NP/Ti₃C₂T_x ($+0.18$ eV). Therefore, the greater adsorption energy absolute value of Cu-SA/Ti₃C₂T_x led to a stable absorption regime, which accounts for the suppression of the HER activity in Cu-SA/Ti₃C₂T_x in comparison with the Cu-SA/Ti₃C₂T_x counterpart. This finding is also consistent with the experimental results (Figure 2b,c). In addition to Supplementary Figure 41, the description of the above content has been added in Page 15 of the revised main text file.

6. Another consideration into the different performance between the single atom and nanoparticles could be due to solvation. Explicit solvation is commonly included in computational studies for CO₂RR or approximated with energetic corrections. Due to the different local geometry of the single atom site the solvation shell and stabilization energies will likely differ. This should at least be addressed in text or via calculations as a possible explanation.

Response:

Thank you very much for this helpful suggestion. Indeed, the solvent effect is a critical factor that may affect the conformations of intermediate adsorbates and the accuracy of free energy. By considering the solvating waters, the explicit solvent

method gives a more intuitively realistic picture of the practical reaction that occurred in the aqueous phase.

In this work, given that our $\text{Ti}_3\text{C}_2\text{T}_x$ substrate model surface is OH-terminated, and at the same time, the reaction adsorbates are different in each elemental reaction step, therefore, rather different conformations will be produced by reacting with a large number of solvent molecules after consideration of the explicit solvation effect. Therefore, the full analysis of the energetics of the complex pathways towards C_{2+} products after explicit consideration solvent layers would need a lot of time and cost. However, restricted by our shoestring computational budget, the number and configurations of water molecules included in the explicit solvation model are quite limited, leading to results lack of generality.

Here, the implicit solvation model is a feasible compromise for this issue, which represents solvent as a continuous medium instead of individual “explicit” solvent molecules. It need not consider the complex real solvent molecules, showing an advantage of fast convergence. Previous studies have shown that the implicit model can provide some insights into CO_2 reduction (*J. Am. Chem. Soc.* 2017, 139, 130; *J. Am. Chem. Soc.* 2016, 138, 483; *Chem* 2017, 3, 652; *Nat. Commun.* 2018, 9, 1320). Therefore, the implicit solvation model is used here using the Poisson–Boltzmann model implemented in VASPsol (*Phys. Rev. B* 2012, 86, 075140), and the calculated results have been added in newly drawn Figure 3a,b in Page 13 of the revised main text file.

The computational parameters and details have been added in Pages 20 and 21 of the revised main text file. The paper (*Phys. Rev. B* 2012, 86, 075140) has been cited as ref. 50 in the reference list of the revised main text file.

7. For free energy calculations (G), were ZPE and entropy contributions (at reaction

temperature) included? They should be, and if so, should be mentioned in the computational method.

Response:

Thank you very much for your valuable comments. In this revision, all of the mentioned energy contributions, including ZPE and entropy contributions (at 300 K), were applied and described in the “computation details” section in Pages 20–22 of the revised main text file. The energy corrections for the different adsorbates have been added to Supplementary Tables 3 and 4 in Pages 47 and 48 in the revised SI file, respectively.

8. Pg 13 Line 210 "mechanical" should be mechanistic.

Response:

Thank you for pointing this out. We are so sorry for the mistake we made. The typo error has been corrected. We carefully checked and confirmed that no similar mistakes exist in the revised main text and SI files.

Reviewers' Comments:

Reviewer #2:

Remarks to the Author:

After carefully check the revised manuscript, I found that the authors have made significant improvements and corrected all the problems the referee proposed. So, I recommend to publish the paper as it is.

Reviewer #3:

Remarks to the Author:

Although the authors made a major revision based on the reviewers' comments, the structure of catalyst and the identification of the catalytic active sites still remain ambiguous. Therefore, the reviewer cannot recommend the acceptance of this work in Nature Communications.

1: The coordination environment of Cu-SA is unclear, Cu atom coordinated with 3 O atoms or 3 O and 3 Ti atoms? The authors claimed the Bader charge of Cu in Cu-SA/Ti₃C₂T_x to be about +0.42, which model is used to calculate the Bader charge?

2: Experimental data is required to show that the isolated Cu atom is the real active site for catalytic CO reduction.

3: In Figure 3d, the Cu atom is connected to 2 O atoms, and all of the DFT calculations were conducted based on this model; it is hard to connect the structure characterization with the theoretical study, making the theoretical study not meaningful.

4: Something wrong in caption of Figure 3, "d, e Charge density difference of the *COCO-adsorbed and *COCHO-adsorbed configuration in e Cu (111) and f Cu-SA/Ti₃C₂T_x, respectively."

5: The authors were asked to provide an example NMR spectrum to demonstrate and illustrate the analysis of the liquid products? In the revised version of the manuscript, one NMR spectrum was provided as SI Fig. 23c, however, this NMR spectrum cannot provide any quantitative information of the product concentration. The authors described that "The ¹H NMR analysis of liquid products when the electrolysis experiment was conducted under an open-circuit voltage." Why conducting the experiment at the open-circuit voltage?

6: In the DFT calculations, why is the *CHO + CO → *CHO-CO pathway energetically more favorable for Cu-SA/Ti₃C₂T_x, as compared to the case on Cu (111) surface? or 2*CO → *COCO on the Cu-SA/Ti₃C₂T_x?

Reviewer #4:

Remarks to the Author:

The authors have made very thorough revisions in response to the comments. Overall, the revised results are very impressive and significantly help to improve the theoretical understanding of this catalyst system.

Specifically, they have provided ample evidence for the model system they use, and provided quantification(s) of their stability (formation energy, diffusion barriers, charge state, etc.). The reaction profile is also much more complete and supplemented by reaction barriers. The lower C-C coupling barrier on the single atom system is also enlightening to see. The improved catalytic performance of the modeled single atom system over the Cu (111) surface appears to be convincing.

I don't believe additional revisions are necessary from the theoretical side.

Responses to the Comments

Responses to the comments of Reviewer #2:

Comments:

After carefully check the revised manuscript, I found that the authors have made significant improvements and corrected all the problems the referee proposed. So, I recommend to publish the paper as it is.

Response:

Thank you very much for reviewing our revised manuscript. We greatly appreciate your inspiring and constructive comments.

Responses to the comments of Reviewer #3:

Comments:

Although the authors made a major revision based on the reviewers' comments, the structure of catalyst and the identification of the catalytic active sites still remain ambiguous. Therefore, the reviewer cannot recommend the acceptance of this work in Nature Communications.

Response:

We truly thank you for reviewing the revised version of our manuscript and greatly appreciate your helpful and constructive comments. Details of the corresponding answers and changes made are described below point by point. For your convenience, all changes have been highlighted in yellow in the revised main text and the revised Supplementary Information (SI) files.

1: The coordination environment of Cu-SA is unclear, Cu atom coordinated with 3 O atoms or 3 O and 3 Ti atoms? The authors claimed the Bader charge of Cu in Cu-SA/Ti₃C₂T_x to be about +0.42, which model is used to calculate the Bader charge?

Response:

Thank you very much for pointing this out. Actually, Cu atom was coordinated with 3 O atoms in the first coordination shell and 3 Ti atoms in the second coordination shell as illustrated by the EXAFS fittings. That is to say, the Cu atom was chemically bonded with 3 nearest O atoms, while the farther 3 Ti atoms also influenced the electronic structure of Cu atoms, which contribute to the peak at 2.7 Å in FT-EXAFS spectrum (Figure R1a). The distances of Cu atoms to both shells are consistent with our fitting models (Figure 1g). To better illustrate this, Figure R1b,c shows the schemes of coordination shell of one Cu single atom in Cu-SA/Ti₃C₂T_x.

Figure R1. (a) The first two-shell (O, Ti) fittings of the FT-EXAFS spectrum for Cu-SA/Ti₃C₂T_x. (b, c) Schemes of coordination shells around Cu SAs. Note that Figure R1a is the same one as Supplementary Figure 11 in the previously revised version.

Figure R2. Bader charge in Cu-SA/Ti₃C₂T_x model.

Regarding your second point, the Bader charge of Cu in Cu-SA/Ti₃C₂T_x is calculated using the optimized DFT model (the results are added as Figure R2), which is

consistent with the model used in EXAFS fitting and theoretical calculations.

The description of the above content has been added in Page 7 of the revised main text file. Figures R1 and R2 have been added as Supplementary Figures 11 and 12 in Pages 12 and 13 of the revised SI file, respectively.

2: Experimental data is required to show that the isolated Cu atom is the real active site for catalytic CO reduction.

Response:

Thank you for the constructive comments. In the last submitted version, the comprehensive characterizations of Cu-SA/Ti₃C₂T_x, including atomic-resolution high-angle annular dark-field scanning transmission electron microscopy (HAADF-STEM) images and X-ray absorption fine structure (XAFS), have proved that the stable presence of Cu single atoms on Ti₃C₂T_x. In addition, to validate the high reactivity of Cu-SA/Ti₃C₂T_x, a series of control samples (i.e. Cu-NP/Ti₃C₂T_x, Ti₃C₂T_x and R-Ti₃C₂T_x) have been conducted. Cu-NP/Ti₃C₂T_x shows a very low current density, and the highest Faradic efficiencies (FEs) of C₂H₄ and EtOH are about 3.4- and 2.1-fold lower than that on Cu-SA/Ti₃C₂T_x, respectively (Figure 2b). This circumstance was not improved even increasing the Cu content in Cu-NP/Ti₃C₂T_x from 5.2 to 20.3 wt%. For pristine Ti₃C₂T_x and reduced Ti₃C₂T_x (i.e. R-Ti₃C₂T_x), the maximum C₂H₄ FEs of Ti₃C₂T_x and R-Ti₃C₂T_x are significantly lower than that of Cu-SA/Ti₃C₂T_x, demonstrating that the Ti₃C₂T_x support helps to capture and stabilize Cu species and the impact on the selectivity of CO reduction is negligible. All the control results indicated that the high CO reduction activity of Cu-SA/Ti₃C₂T_x comes from the Cu single atoms on Ti₃C₂T_x. Those experiment results are also in line with the DFT calculation results, in which Cu-SA/Ti₃C₂T_x showed much lower free energy barriers for CO reduction than that of Cu (111).

According to your comments, additional control experiments have been conducted to further prove the reactivity of Cu single-atom sites for catalytic CO reduction. Since that the SCN^- anions can coordinate with Cu atoms and poison the Cu single sites during the catalysis (Small 2019, 15, 1902410; iScience 2019, 22, 97). We therefore have examined CO reduction activity of Cu-SA/ $\text{Ti}_3\text{C}_2\text{T}_x$ in 1 M KOH electrolyte containing 0.1 mM KSCN. The corresponding CO electroreduction results are added as Figure R3. In the presence of SCN^- , Cu-SA/ $\text{Ti}_3\text{C}_2\text{T}_x$ exhibits a noticeable reduction in current density, meanwhile, the obtained highest C_{2+} Faradic efficiency of Cu-SA/ $\text{Ti}_3\text{C}_2\text{T}_x$ is as low as 26% (-0.7 V vs. RHE), much smaller than that of electrolyte without SCN^- (Figure 2a,b). These results demonstrate that Cu single atoms act as the CO reduction sites.

Figure R3. SCN^- poisoning in CO-saturated 1 M KOH. (a) LCV curves of Cu-SA/ $\text{Ti}_3\text{C}_2\text{T}_x$ in different electrolyte solutions, of which the black line is the same as the red one in Figure 2a. (b) FEs of Cu-SA/ $\text{Ti}_3\text{C}_2\text{T}_x$ in CO-saturated 1 M KOH + 0.1 mM KSCN at different applied potentials.

In addition, the COR pathway on pure $\text{Ti}_3\text{C}_2\text{T}_x$ was also calculated by the DFT method and the results are added as Figure R4. Note that the $2^*\text{CO} \rightarrow ^*\text{COCO}$ pathway is infeasible on Cu-SA/ $\text{Ti}_3\text{C}_2\text{T}_x$ model, and the $^*\text{CHO} + \text{CO} \rightarrow ^*\text{CHO-CO}$ pathway is adopted here. The results show that a large energy barrier of 1.29 eV is required for the activation of CO on $\text{Ti}_3\text{C}_2\text{T}_x$, much higher than that on Cu-SA/ $\text{Ti}_3\text{C}_2\text{T}_x$. This agrees

well with the poor COR performance of $\text{Ti}_3\text{C}_2\text{T}_x$. It further confirmed the high COR activity of $\text{Cu-SA}/\text{Ti}_3\text{C}_2\text{T}_x$ comes from the Cu single atoms on $\text{Ti}_3\text{C}_2\text{T}_x$.

Figure R4. The optimized COR pathway on pure $\text{Ti}_3\text{C}_2\text{T}_x$ surface.

The description of the above content has been added in Pages 11 and 13 of the revised main text file. Figures R3 and R4 have been added as Supplementary Figures 35 and 45 in Pages 36 and 46 of the revised SI file, respectively. The papers (Small 2019, 15, 1902410; iScience 2019, 22, 97) have been cited as refs. 48 and 49 in the reference list of the main text file, respectively.

3: In Figure 3d, the Cu atom is connected to 2 O atoms, and all of the DFT calculations were conducted based on this model; it is hard to connect the structure characterization with the theoretical study, making the theoretical study not meaningful.

Response:

Thank you for your insightful question. In this revised version, all the DFT calculations were conducted using Cu-O_3 as the initial model. However, when an intermediate was on absorbed Cu single atoms, the optimization process was proceeded to obtain

the optimal adsorption configuration. In some adsorption configurations in Figure 3c, one Cu–O bond was elongated to get the lowest energy state. For example, for *COCHO-adsorbed Cu-SA/Ti₃C₂T_x (Figure 3d), one of the Cu–O bonds was elongated to ~2.9 Å, and it was not displayed since the weakening interaction between Cu and O. Actually, there is still electron exchange between Cu atoms and O atoms, as illustrated in the charge density difference maps Figure 3d. The dynamic changes of single atoms during the reaction were also reported by Fang et al. in Nat. Comm. 2020, 11, 1029. The description of the above content has been added in the caption of Figure 3 in Page 14 of the revised main text file.

4: Something wrong in caption of Figure 3, “d, e Charge density difference of the *COCO-adsorbed and *COCHO-adsorbed configuration in e Cu (111) and f Cu-SA/Ti₃C₂T_x, respectively.”

Response:

Thank you very much for pointing this out. We are so sorry for the mistakes we made. These errors have been corrected. Meantime, we have carefully checked and confirmed that no similar mistakes exist in the revised main text and SI files.

5: The authors were asked to provide an example NMR spectrum to demonstrate and illustrate the analysis of the liquid products? In the revised version of the manuscript, one NMR spectrum was provided as SI Figure 23c, however, this NMR spectrum cannot provide any quantitative information of the product concentration. The authors described that “The ¹H NMR analysis of liquid products when the electrolysis experiment was conducted under an open-circuit voltage.” Why conducting the experiment at the open-circuit voltage?

Response:

Thanks for your helpful comments. As you stated, the ^1H NMR spectrum in Supplementary Figure 23c was conducted under an open-circuit voltage, which was given as evidence for that: the EtOH was generated from the electroreduction of dissolved CO by Cu-SA/Ti₃C₂T_x rather than the residual EtOH. In the last revised version, we have added the standard curves (Figure R4a,b), which were made using standard chemicals over the concentration range of interest (ethanol and acetate), with the internal standard DMSO in 1 M KOH. The description for quantifying the concentrations of ethanol and acetate collected at -0.7 V vs RHE for 2 h was also given in Page 14 of the last submitted SI file (Page 15 in the current version).

Figure R4. (a-b) Standard curves for EtOH (a) and acetate (b) products for ^1H NMR analysis. (c) ^1H NMR spectra of the electrolyte taken after 2-h electrocatalysis over Cu-SA/Ti₃C₂T_x at -0.7 V (RHE). Note that Figure R4a,b is the same one of Supplementary Figure 14a,b in the previously revised version.

According to your suggestion, the representative ^1H NMR pattern recorded at -0.7 V vs. RHE for 2 h is added as Figure R4c. The ratios of the areas of the produced acetate

and EtOH (peak 1) to the DMSO peak area were calculated to be 27.98 and 0.05164, respectively. The obtained ratios were then compared to the standard curves (Figure R4a,b) to quantify the concentrations of the reaction products. Accordingly, the concentrations of EtOH and acetate (the blue balls in Figure R4a,b) were measured to be 5.4 and 0.88 mM, respectively. In this case, the corresponding formation rates of EtOH and acetate were calculated to be 2.79 and 0.44 mM h⁻¹ at -0.7 V vs. RHE, respectively.

The description of the above content has been added in Page 15 of the revised SI file. Figure R4c has been added as Supplementary Figure 14c in the revised SI file.

6: In the DFT calculations, why is the *CHO + CO → *CHO-CO pathway energetically more favorable for Cu-SA/Ti₃C₂T_x, as compared to the case on Cu (111) surface? or 2*CO → *COCO on the Cu-SA/Ti₃C₂T_x?

Response:

Thank you very much for the very helpful instructions. Actually, both *CHO + CO → *CHO-CO and 2*CO → *COCO coupling processes probably existed in CO reductions (ACS Catal. 2018, 8, 1490, ref. 56). The 2*CO → *COCO process is always reported on Cu bulks, which needs two nearest Cu adsorption sites. According to your suggestion, the 2*CO → *COCO process was also calculated on Cu-SA/Ti₃C₂T_x. A model of 2Cu/Ti₃C₂T_x was used, in which two nearest isolated Cu atoms (a distance of 3.2 Å) were constructed on Ti₃C₂T_x substrate (Figure R5). Note that, the distance is nearer than the actual average Cu-Cu interatomic distance (0.61 Å) in Cu-SA/Ti₃C₂T_x (Supplementary Figure 6), and this 2Cu/Ti₃C₂T_x model was just used for investigating the possibility of 2*CO coupling on Cu-SA/Ti₃C₂T_x. Accordingly, the geometric optimization was proceeded on *COCO-2Cu/Ti₃C₂T_x model to investigate the existence of *COCO intermediate. Unfortunately, the energy minimum value point for *COCO intermediate was not found in the local region. Instead, *COCO was

separated to form two isolated *CO (Figure R5). So the $2^*CO \rightarrow ^*COCO$ pathway is infeasible on the $2Cu/Ti_3C_2T_x$ surface. Therefore, the $^*CHO + CO \rightarrow ^*CHO-CO$ pathway is adopted for the calculation of CO reduction pathways of $Cu-SA/Ti_3C_2T_x$.

Figure R5. The optimization process of *COCO on $Cu-SA/Ti_3C_2T_x$.

In addition, the $^*CHO + CO \rightarrow ^*CHO-CO$ pathway is also calculated on $Cu(111)$. As illustrated in Figure R6, the $^*CO \rightarrow ^*CHO$ becomes the rate-limiting step with a free energy barrier of 1.18 eV, a little higher than that of $2^*CO \rightarrow ^*COCO$ step (0.94 eV) on $Cu(111)$. Therefore, the two pathways co-existed on $Cu(111)$, however, they are both energetically more unfavorable than the $^*CHO + CO \rightarrow ^*CHO-CO$ pathway on $Cu-SA/Ti_3C_2T_x$, suggesting the inferior CO reduction activity of $Cu(111)$.

Figure R6. The reaction mechanism of the COR on $Cu(111)$ through *CHO pathway.

The description of the above content has been added in Page 13 of the revised main text file and Page 42 in the revised SI file, respectively. Figures R5 and R6 have been added as Supplementary Figures 41 and 42 in Pages 42 and 43 of the revised SI file, respectively.

Responses to the comments of Reviewer #4:

Comments:

The authors have made very thorough revisions in response to the comments. Overall, the revised results are very impressive and significantly help to improve the theoretical understanding of this catalyst system.

Specifically, they have provided ample evidence for the model system they use, and provided quantification(s) of their stability (formation energy, diffusion barriers, charge state, etc.). The reaction profile is also much more complete and supplemented by reaction barriers. The lower C-C coupling barrier on the single atom system is also enlightening to see. The improved catalytic performance of the modeled single atom system over the Cu (111) surface appears to be convincing.

I don't believe additional revisions are necessary from the theoretical side.

Response:

We are very grateful to your encouraging and positive comments and really appreciate your agreement of acceptance with this revised manuscript.

Reviewers' Comments:

Reviewer #3:

Remarks to the Author:

The structure of the catalyst and identification of the catalytic sites still remain ambiguous. Therefore, the reviewer does not believe the manuscript is suitable for Nature Communications.

1. The claimed real Cu-O3 active site seems not "real" in the CO electroreduction process. There is no characteristic signal attributed to the adsorption of CO on Cu SAC resulted from CO drift experiments (Supplementary Fig. 13), suggesting weak CO adsorption on the surface of Cu-SA/Ti3C2Tx. Therefore, there is no direct evidence to confirm the CO reduction reaction by the SAC-Cu site. In addition, previous experimental results reveal that the in-situ generated Cu cluster from the single atom Cu catalyst is the real active site(s), which can also recover to single atom sites after catalyzing CO2 reduction to multi-carbon products (Angew. Chem. Int. Ed. 2019, 58, 15098-15103). Although the Cu SAC was well-preserved after electrochemical testing, the proposed Cu-O3 active site is not convincing and the DFT analysis based on the Cu-O3 structure is unbelievable.
2. The HAADF-STEM images exhibit very bright Cu dots on the Ti-based support (Fig. 1d)!!! It is unreasonable due to the small difference in Z contrast of Cu and Ti element. The authors should make a reasonable explanation.
3. For presenting the XPS data (such as, Supplementary Fig. 9 and 18), generally, the survey spectra should be presented together with high resolution spectra for each element. Moreover, a comparison of chemical state for each element should be conducted between Cu-SA/Ti3C2Tx and pure Ti3C2Tx.
4. What's more, the mass spectroscopy data (Supplementary Fig. 23.) deserve the authors' careful consideration. Why does the CH3CH2O- fragment not show up in the fragments of C2H5OH? Generally, there should be lots of CH3CH2O- in the ethanol ion fragments. Obviously, this fragment does not exist in the experimental data given by the authors either in the non-isotopically labeled CO2 reduction experiment or in the isotopically labeled CO2 experiment.

Responses to the Comments

Responses to the comments of Reviewer #3:

Comments:

The structure of the catalyst and identification of the catalytic sites still remain ambiguous. Therefore, the reviewer does not believe the manuscript is suitable for Nature Communications.

Response:

We thank you for reviewing the revised version of our manuscript and greatly appreciate your helpful and constructive comments. Details of the corresponding answers and changes made are described below point by point. For your convenience, all changes have been highlighted in yellow in the revised main text and the revised supplementary information (SI) files.

1: The claimed real Cu-O₃ active site seems not “real” in the CO electroreduction process. There is no characteristic signal attributed to the adsorption of CO on Cu SAC resulted from CO drift experiments (Supplementary Fig. 13), suggesting weak CO adsorption on the surface of Cu-SA/Ti₃C₂T_x. Therefore, there is no direct evidence to confirm the CO reduction reaction by the SAC-Cu site. In addition, previous experimental results reveal that the in-situ generated Cu cluster from the single atom Cu catalyst is the real active site(s), which can also recover to single atom sites after catalyzing CO₂ reduction to multi-carbon products (Angew. Chem. Int. Ed. 2019, 58, 15098-15103). Although the Cu SAC was well-preserved after electrochemical testing, the proposed Cu-O₃ active site is not convincing and the DFT analysis based on the Cu-O₃ structure is unbelievable?

Response:

Thank you for pointing this out. In this work, the CO reduction reaction by Cu single

sites has been proved by the rigorous control experiments, including pure $\text{Ti}_3\text{C}_2\text{T}_x$, $\text{R-Ti}_3\text{C}_2\text{T}_x$, $\text{Cu-NP/Ti}_3\text{C}_2\text{T}_x$ control samples (Supplementary Fig. 32) and SCN^- poison experiment (Supplementary Fig. 36). While for the CO DRIFTS experiments in Supplementary Fig. 13, the experiment condition is rather different from that of the CO electroreductions. For CO-DRIFTS spectra tests, they were conducted under a CO/He (5 vol%) gas atmosphere without an applied potential. For CO electroreduction reactions, the CO directly follows into the electrolytic cell with water providing protons and electricity providing electrons. Therefore, CO-DRIFTS test may be an effective method for determining the existence of single atoms for a few noble metals, but it absolutely can not reflect the CO adsorption on catalyst in practical CO electroreduction reactions.

For your another concern about the previous experimental reports, they indeed show the reversible formation of Cu clusters from Cu single atoms under CO reduction using the in situ X-ray absorption structure technology. But unfortunately, we have no chance to conduct this test. However, we noticed that the structure of our Cu single atoms is rather different from the previous report (Angew. Chem. Int. Ed. 2019, 58, 15098–15103). Unlike the previous reported Cu-N_x sites, the Cu single atoms on $\text{Ti}_3\text{C}_2\text{T}_x$ are difficult to move away from the oxygen traps or forming a Cu dimer due to the high binding energy and migration barriers of Cu single atoms, as we previously stated in Supplementary Figs. 20–21 of the manuscript. Therefore, the Cu single atoms on $\text{Ti}_3\text{C}_2\text{T}_x$ are stable during the CO reduction test, which also agrees with the HAADF-STEM, XRD and XPS characterizations (Supplementary Figs. 17–19) of $\text{Cu-SA/Ti}_3\text{C}_2\text{T}_x$ after the electrochemical test. Furthermore, a possible dimer model of $2\text{Cu/Ti}_3\text{C}_2\text{T}_x$ was also used to calculate the $2^*\text{CO} \rightarrow ^*\text{COCO}$ process on $\text{Cu-SA/Ti}_3\text{C}_2\text{T}_x$, but it is proved to be infeasible during the optimization process (Supplementary Fig. 47). This result also confirmed the single Cu atoms as active sites for CO reduction.

For your question about the structure of Cu-O_3 sites on $\text{Ti}_3\text{C}_2\text{T}_x$, we believe that there

are enough convincing results for the determination of the structure, including the atomic resolution HAADF images, simulation images and the X-ray absorption fine structure test and fittings. Therefore, the detailed and rigorous DFT analysis based on this atomic Cu-O₃ structures is also believable. Furthermore, the DFT analysis has also been synthetically revised and improved according to the suggestions of reviewer #4. The DFT results are also consistent with the experiments. Therefore, the good catalytic performance of the modeled single atom system can be well explained by the DFT results.

2: The HAADF-STEM images exhibit very bright Cu dots on the Ti-based support (Fig. 1d)!!! It is unreasonable due to the small difference in Z contrast of Cu and Ti element. The authors should make a reasonable explanation.

Response:

Thank you for your comment. Indeed, the HAADF-STEM image intensity is largely dependent on the atomic number (Z) of the element, which is approximately proportional to $Z^{1.7}$. The bigger difference between Z numbers are, the more obvious the image contrast is. For Cu-SA/Ti₃C₂T_x, the atomic number of Cu and Ti is 29 and 22, respectively. In theory, they can be discerned because of enough gap between the atomic numbers. In addition, other factors, such as the sample thickness, defocus values, electron channeling effects, the spacial position of atoms, debye-waller factors, cross-talk effects and so on, also influence the image intensities. For Cu-SA/Ti₃C₂T_x, the well-arrangement of Ti atoms, 2-dimensional ultrathin structure and the flat surface of Ti₃C₂T_x all lead to a better view for the attached Cu atoms.

According to your suggestion, we also simulated the HAADF-STEM image of Cu-SA/Ti₃C₂T_x using the structural model proposed in our manuscript. The frozen phone approximated multislice simulations were used with the QSTEM program. As shown in Figure R1, the intensity of the simulated image matched well with the experimental ones. Therefore, we confirmed that the bright spots on Ti₃C₂T_x are Cu

single atoms.

Figure R1 (a, c) The experimental atomic resolution HAADF-STEM image and the corresponding simulated image. (b) The image intensity of the experimental HAADF-STEM image and the corresponding simulated images. (d) The DFT-optimized Cu-SA/Ti₃C₂T_x structure.

The description of the above content has been added in Page 6 of the revised SI file.

Figures R1 has been added as Supplementary Fig. 5 of the revised SI file.

3: For presenting the XPS data (such as, Supplementary Fig. 9 and 18), generally, the survey spectra should be presented together with high resolution spectra for each element. Moreover, a comparison of chemical state for each element should be conducted between Cu-SA/Ti₃C₂T_x and pure Ti₃C₂T_x.

Response:

Thank you for pointing it out. According to your suggestion, the survey spectra and

the high-resolution spectra for each element of Cu-SA/Ti₃C₂T_x (including those of after stability tests) and pure Ti₃C₂T_x have been added in Figures R2 and R3.

A comparison of the chemical state for each element in Cu-SA/Ti₃C₂T_x and pure Ti₃C₂T_x have been conducted. As shown in Figure R2, Cu 2p peaks appeared at 932.5 and 952.4 eV for Cu-SA/Ti₃C₂T_x, suggesting the successful incorporation of Cu atoms. High-resolution Ti 2p XPS spectra show two peaks at around 455.5 and 461.3 eV, corresponding to Ti interactions with carbons and terminal atoms (i.e. C-Ti-T_x). The Ti⁴⁺ at 458.9 eV corresponds to Ti-O bonds on terminals. No obvious increase in Ti⁴⁺ state can be observed, indicating the well-preserved of Ti state in Cu-SA/Ti₃C₂T_x. The high-resolution C 1s XPS spectra show two peaks at 281.9 and 284.7 eV, corresponding to the C-Ti and C-C bonds. After the introduction of Cu atoms, no shift of C-C peak can be noticed, indicating no obvious interaction existed between Cu and C atoms. For high-resolution O 1s XPS spectra, the Ti-O, C-Ti-O and C-Ti-OH bonds can be fitted at 529.7, 530.8 and 532.0 eV, respectively. The ratio of C-Ti-O to C-Ti-OH bond increased from 17.6% to 28.2% after the introduction of Cu, agree with the formation of Cu single atoms in Ti₃C₂T_x.

Figure R2. XPS survey and high-resolution XPS characterization of Cu-SA/Ti₃C₂T_x and pure Ti₃C₂T_x. (a) XPS survey, (b) C 1s XPS spectra, (c) O 1s XPS spectra, (d) Ti 2p XPS spectra, and (e) Cu 2p XPS spectra.

Figure R3. High-magnification XPS spectra of Cu 2p for Cu-SA/Ti₃C₂T_x after stability test. (a) XPS survey, (b) Cu 2p XPS spectrum, (c) O 1s XPS spectrum, (d) C 1s XPS spectrum, (e) Ti 2p XPS spectrum. No obvious change can be observed compared with that of fresh Cu-SA/Ti₃C₂T_x (Figure R2), indicating the chemical stability of the Cu SAs.

Figures R2 and R3 have been added as Supplementary Figs. 10 and 19 in the revised SI file. The description of the above content has been added in Pages 11–12, 21 of

the revised SI file.

4: What's more, the mass spectroscopy data (Supplementary Fig. 23.) deserve the authors' careful consideration. Why does the $\text{CH}_3\text{CH}_2\text{O}^-$ fragment not show up in the fragments of $\text{C}_2\text{H}_5\text{OH}$? Generally, there should be lots of $\text{CH}_3\text{CH}_2\text{O}^-$ in the ethanol ion fragments. Obviously, this fragment does not exist in the experimental data given by the authors either in the non-isotopically labeled CO_2 reduction experiment or in the isotopically labeled CO_2 experiment.

Response:

Thank you for your comment. In the last revised version, Supplementary Fig. 23 represents the mass spectra of acetic acid, rather than the $\text{C}_2\text{H}_5\text{OH}$ (EtOH). The mass spectra of $\text{C}_2\text{H}_5\text{OH}$ was given as Supplementary Fig. 22 in the initially submitted SI file (it has been renumbered as Supplementary Fig. 23 in the revised version because of the addition of a new figure), in which lots of $\text{CH}_3\text{CH}_2\text{O}^-$ fragments have shown up at $m/z = 45$ (Figure R4). The fingerprint feature of the $\text{CH}_3\text{CH}_2\text{OH}$ mass spectra is also consistent with previous reports (ref. 47). For a better view, all the main peaks of the mass spectra have been noted in the revised Supplementary Figs. 22–24 of the SI file.

Figure R4. Typical mass spectra of the EtOH standard (a) and ^{13}C -EtOH sample (b).

Note that the signal at $m/z = 47$ in (a) is due to the natural abundance of ^{13}C .